# Bridging Discrete and Backpropagation: Straight-Through and Beyond

**Liyuan Liu   Chengyu Dong   Xiaodong Liu   Bin Yu   Jianfeng Gao**
Microsoft Research
{lucliu, v-chedong, xiaodl, v-ybi, jfgao}@microsoft.com

## Abstract

Backpropagation, the cornerstone of deep learning, is limited to computing gradients for continuous variables. This limitation poses challenges for problems involving discrete latent variables. To address this issue, we propose a novel approach to approximate the gradient of parameters involved in generating discrete latent variables. First, we examine the widely used Straight-Through (ST) heuristic and demonstrate that it works as a first-order approximation of the gradient. Guided by our findings, we propose ReinMax, which achieves second-order accuracy by integrating Heun's method, a second-order numerical method for solving ODEs. ReinMax does not require Hessian or other second-order derivatives, thus having negligible computation overheads. Extensive experimental results on various tasks demonstrate the superiority of ReinMax over the state of the art.

## 1   Introduction

There has been a persistent pursuit to build neural network models with discrete or sparse variables (Neal, 1992). However, backpropagation (Rumelhari et al., 1986), the cornerstone of deep learning, is restricted to computing gradients for continuous variables. Correspondingly, many attempts have been made to approximate the gradient of parameters that are used to generate discrete variables, and most of them are based on the Straight-Through (ST) technique (Bengio et al., 2013).

The development of ST is based on the simple intuition that non-differentiable functions (e.g., sampling of discrete latent variables) can be approximated with the identity function in the backpropagation (Rosenblatt, 1957; Bengio et al., 2013). Due to the lack of theoretical underpinnings, there is neither guarantee that ST can be viewed as an approximation of the gradient, nor guidance on hyper-parameter configurations or future algorithm development. Thus, researchers have to develop different ST variants for different applications in a trial-and-error manner, which is laborious and time-consuming (van den Oord et al., 2017; Liu et al., 2019; Fedus et al., 2021). To address these limitations, we aim to explore *how ST approximates the gradient and how it can be improved.*

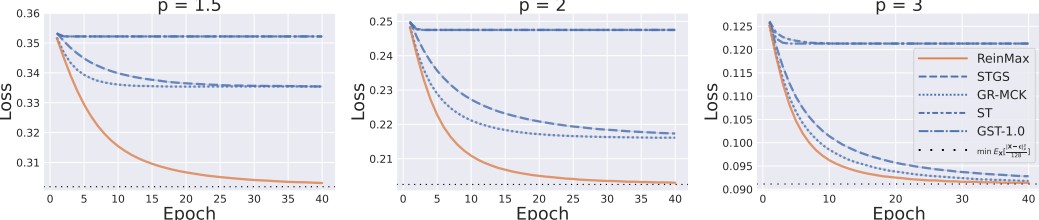

Figure 1: Training curves of polynomial programming, i.e., $\min_{\boldsymbol{\theta}} E_{\boldsymbol{X}}[\|\boldsymbol{X} - \boldsymbol{c}\|_p^p / 128]$, where $\boldsymbol{\theta} \in \mathbb{R}^{128 \times 2}$, $\boldsymbol{X} \in \{0, 1\}^{128}$, and $\boldsymbol{X}_i \overset{\text{iid}}{\sim}$ Multinomial(softmax($\boldsymbol{\theta}_i$)). Details are elaborated in Section 6.

37th Conference on Neural Information Processing Systems (NeurIPS 2023).

| **Algorithm 1:** ST. | **Algorithm 2:** ReinMax. |
|---|---|
| **Input:** $\boldsymbol{\theta}$: softmax input, $\tau$: temperature. | **Input:** $\boldsymbol{\theta}$: softmax input, $\tau$: temperature. |
| **Output:** $\boldsymbol{D}$: one-hot samples. | **Output:** $\boldsymbol{D}$: one-hot samples. |
| 1 $\boldsymbol{\pi}_0 \leftarrow \text{softmax}(\boldsymbol{\theta})$ | 1 $\boldsymbol{\pi}_0 \leftarrow \text{softmax}(\boldsymbol{\theta})$ |
| 2 $\boldsymbol{D} \leftarrow \text{sample\_one\_hot}(\boldsymbol{\pi}_0)$ | 2 $\boldsymbol{D} \leftarrow \text{sample\_one\_hot}(\boldsymbol{\pi}_0)$ |
| 3 $\boldsymbol{\pi}_1 \leftarrow \text{softmax}_\tau(\boldsymbol{\theta})$ | 3 $\boldsymbol{\pi}_1 \leftarrow \frac{\boldsymbol{D}+\text{softmax}_\tau(\boldsymbol{\theta})}{2}$ |
| /* stop_gradient($\cdot$) duplicates its input and detaches it from backpropagation. */ | 4 $\boldsymbol{\pi}_1 \leftarrow \text{softmax}(\text{stop\_gradient}(\ln(\boldsymbol{\pi}_1) - \boldsymbol{\theta}) + \boldsymbol{\theta})$ |
| | 5 $\boldsymbol{\pi}_2 \leftarrow 2 \cdot \boldsymbol{\pi}_1 - \frac{1}{2} \cdot \boldsymbol{\pi}_0$ |
| 4 $\boldsymbol{D} \leftarrow \boldsymbol{\pi}_1 - \text{stop\_gradient}(\boldsymbol{\pi}_1) + \boldsymbol{D}$ | 6 $\boldsymbol{D} \leftarrow \boldsymbol{\pi}_2 - \text{stop\_gradient}(\boldsymbol{\pi}_2) + \boldsymbol{D}$ |
| 5 **return** $\boldsymbol{D}$ | 7 **return** $\boldsymbol{D}$ |

First, we adopt a novel perspective to examine ST and show that it works as a special case of the forward Euler method, approximating the gradient with first-order accuracy. Besides confirming that ST is indeed an approximation of the gradient, our finding provides guidance on how to optimize hyper-parameters of ST and its variants, i.e., ST prefers to set the temperature $\tau \geq 1$, and Straight-Through Gumbel-Softmax (STGS; Jang et al., 2017) prefers to set the temperature $\tau \leq 1$.

Our analyses not only shed insights on the underlying mechanism of ST but also lead us to develop a novel gradient estimation method called ReinMax. ReinMax integrates Heun's Method and achieves second-order accuracy, i.e., its approximation matches the Taylor expansion of the gradient to the second order, without requiring the Hessian matrix or other second-order derivatives.

We conduct extensive experiments on polynomial programming Tucker et al. (2017); Grathwohl et al. (2018); Pervez et al. (2020); Paulus et al. (2021), unsupervised generative modeling (Kingma & Welling, 2013), structured output prediction (Nangia & Bowman, 2018), and differentiable neural architecture search (Dong et al., 2020a) to demonstrate that ReinMax brings consistent improvements over the state of the art[1].

Our contributions are two-fold:

- We formally establish that ST works as a first-order approximation to the gradient in the general multinomial case, which provides valuable guidance for future research and applications.

- We propose a novel and sound gradient estimation method ReinMax that achieves second-order accuracy without requiring the Hessian matrix or other second-order derivatives. ReinMax is shown to outperform the previous state-of-the-art methods in extensive experiments.

## 2 Related Work and Preliminary

**Discrete Latent Variables and Gradient Computation.** The idea of incorporating discrete latent variables and neural networks dates back to sigmoid belief network and Helmholtz machines (Williams, 1992; Dayan et al., 1995). To keep things straightforward, we will focus on a simplified scenario. We refer to the tempered softmax as $\text{softmax}_\tau(\boldsymbol{\theta})_i = \frac{\exp(\boldsymbol{\theta}_i/\tau)}{\sum_{j=1}^n \exp(\boldsymbol{\theta}_j/\tau)}$, where $n$ is the number of possible outcomes, $\boldsymbol{\theta} \in \mathcal{R}^{n \times 1}$ is the parameter, and $\tau$ is the temperature[2]. For $i \in [1, \cdots, n]$, we mark its one-hot representation as $\boldsymbol{I}_i \in \mathcal{R}^{n \times 1}$, whose element equals 1 if it is the $i$-th element or equals 0 otherwise. Let $\boldsymbol{D}$ be a discrete random variable and $\boldsymbol{D} \in \{\boldsymbol{I}_1, \cdots, \boldsymbol{I}_n\}$, we assume the distribution of $\boldsymbol{D}$ is parameterized as: $p(\boldsymbol{D} = \boldsymbol{I}_i) = \boldsymbol{\pi}_i = \text{softmax}(\boldsymbol{\theta})_i$, and mark $\text{softmax}_\tau(\boldsymbol{\theta})$ as $\boldsymbol{\pi}^{(\tau)}$. Given a differentiable function $f : \mathcal{R}^n \to \mathcal{R}$, we aim to minimize (note that temperature scaling is not used in the generation of $\boldsymbol{D}$):

$$\min_{\boldsymbol{\theta}} \mathcal{L}(\boldsymbol{\theta}), \quad \text{where } \mathcal{L}(\boldsymbol{\theta}) = E_{\boldsymbol{D} \sim \text{softmax}(\boldsymbol{\theta})}[f(\boldsymbol{D})]. \tag{1}$$

Here, we mark the gradient of $\boldsymbol{\theta}$ as $\nabla$:

$$\nabla := \frac{\partial \mathcal{L}(\boldsymbol{\theta})}{\partial \boldsymbol{\theta}} = \sum_i f(\boldsymbol{I}_i) \frac{d\boldsymbol{\pi}_i}{d\boldsymbol{\theta}}. \tag{2}$$

---

[1]Implementations are available at `https://github.com/microsoft/ReinMax`.

[2]Without specification, the temperature (i.e., $\tau$) is set to 1.

In many applications, it is usually too costly to compute $\nabla$, since it requires the computation of $\{f(\boldsymbol{I}_1), \cdots, f(\boldsymbol{I}_n)\}$ and evaluating $f(\boldsymbol{I}_i)$ is costly for typical deep learning applications. Correspondingly, many efforts have been made to estimate $\nabla$ efficiently.

The $\nabla_{\text{REINFORCE}}$ (Williams, 1992) is unbiased (i.e., $E[\nabla_{\text{REINFORCE}}] = \nabla$) and only requires the distribution of the discrete variable to be differentiable (i.e., no backpropagation through $f$):

$$\nabla_{\text{REINFORCE}} := f(\boldsymbol{D}) \frac{d \log p(\boldsymbol{D})}{d \boldsymbol{\theta}}. \tag{3}$$

Despite the REINFORCE estimator being unbiased, it tends to have prohibitively high variance, especially for networks that have other sources of randomness (i.e., dropout or other independent random variables). Recently, attempts have been made to reduce the variance of REINFORCE (Gu et al., 2016; Tucker et al., 2017; Grathwohl et al., 2018; Shi et al., 2022). Still, it has been found that the REINFORCE-style estimators fail to work well in many real-world applications. Empirical comparisons between ReinMax and REINFORCE-style methods are elaborated in Section 6.5.

**Efficient Gradient Approximation.** In practice, a popular family of estimators is Straight-Through (ST) estimators. They compute the backpropagation "through" a surrogate that treats the non-differentiable function (e.g., the sampling of $\boldsymbol{D}$) as an identity function. The idea of ST originates from the perceptron algorithm (Rosenblatt, 1957; Mullin & Rosenblatt, 1962), which leverages a modified chain rule and utilizes the identity function as the proxy of the original derivative of a binary output function. Bengio et al. (2013) improves this method by using non-linear functions like sigmoid or softmax, and Jang et al. (2017) further incorporates the Gumbel reparameterization. Here, we briefly describe Straight-Through (ST) and Straight-Through Gumbel-Softmax (STGS).

In the general multinomial distribution case, as in Algorithm 1, the ST estimator treats the sampling process of $\boldsymbol{D}$ as an identity function during the backpropagation[3]:

$$\widehat{\nabla}_{\text{ST}} := \frac{\partial f(\boldsymbol{D})}{\partial \boldsymbol{D}} \cdot \frac{d \boldsymbol{\pi}}{d \boldsymbol{\theta}}. \tag{4}$$

In practice, $\widehat{\nabla}_{\text{ST}}$ is usually implemented with the tempered softmax, under the hope that the temperature hyper-parameter $\tau$ may be able to reduce the bias introduced by $\widehat{\nabla}_{\text{ST}}$ (Chung et al., 2017).

The STGS estimator is built upon the Gumbel re-parameterization trick (Maddison et al., 2014; Jang et al., 2017). It is observed that the sampling of $\boldsymbol{D}$ can be reparameterized using Gumbel random variables at the zero-temperature limit of the tempered softmax (Gumbel, 1954):

$$\boldsymbol{D} = \lim_{\tau \to 0} \text{softmax}_\tau(\boldsymbol{\theta} + \boldsymbol{G}) \quad \text{where } \boldsymbol{G}_i \text{ are i.i.d. and } \boldsymbol{G}_i \sim \text{Gumbel}(0, 1).$$

STGS treats the zero-temperature limit as identity function during the backpropagation:

$$\widehat{\nabla}_{\text{STGS}} := \frac{\partial f(\boldsymbol{D})}{\partial \boldsymbol{D}} \cdot \frac{d \, \text{softmax}_\tau(\boldsymbol{\theta} + \boldsymbol{G})}{d \boldsymbol{\theta}}. \tag{5}$$

Both $\widehat{\nabla}_{\text{ST}}$ and $\widehat{\nabla}_{\text{STGS}}$ are clearly biased. However, since the mechanism of ST is unclear, it remains unanswered what the form of their biases are, how to configure their hyper-parameters for optimal performance, or even whether $E[\widehat{\nabla}_{\text{ST}}]$ or $E[\widehat{\nabla}_{\text{STGS}}]$ can be viewed as an approximation of $\nabla$. Thus, we aim to answer the following questions: *How $\widehat{\nabla}_{ST}$ approximates $\nabla$ and how it can be improved?*

## 3 Discrete Variable Gradient Approximation: a Numerical ODE Perspective

In numerical analysis, extensive studies have been conducted to develop numerical methods for solving ordinary differential equations. In this study, we leverage these methods to approximate $\nabla$ with the gradient of $f$. To begin, we demonstrate that ST works as a first-order approximation of $\nabla$. Then, we propose ReinMax, which integrates Heun's method for a better gradient approximation and achieves second-order accuracy.

---

[3]We use the notation $\widehat{\nabla}$ to indicate gradient approximations. Note that the generation of $\boldsymbol{D}$ is not differentiable, and $\widehat{\nabla}_{\text{ST}}$ does not have the term $\partial \boldsymbol{D}/\partial \boldsymbol{\pi}$.

## 3.1 Straight-Through as a First-order Approximation

We start by defining a first-order approximation of $\nabla$ as $\widehat{\nabla}_{\text{1st-order}}$.

**Definition 3.1.** *One first-order approximation of $\nabla$ is* $\widehat{\nabla}_{\text{1st-order}} := \sum_i \sum_j \boldsymbol{\pi}_j \frac{\partial f(\boldsymbol{I}_j)}{\partial \boldsymbol{I}_j}(\boldsymbol{I}_i - \boldsymbol{I}_j)\frac{d\boldsymbol{\pi}_i}{d\boldsymbol{\theta}}$.

To understand why $\widehat{\nabla}_{\text{1st-order}}$ is a first-order approximation, we rewrite $\nabla$ in Equation 2 as[4]:

$$\nabla = \sum_i (f(\boldsymbol{I}_i) - E[f(\boldsymbol{D})])\frac{d\boldsymbol{\pi}_i}{d\boldsymbol{\theta}} + \sum_i E[f(\boldsymbol{D})]\frac{d\boldsymbol{\pi}_i}{d\boldsymbol{\theta}} = \sum_i \sum_j \boldsymbol{\pi}_j(f(\boldsymbol{I}_i) - f(\boldsymbol{I}_j))\frac{d\boldsymbol{\pi}_i}{d\boldsymbol{\theta}}. \quad (6)$$

Comparing $\widehat{\nabla}_{\text{1st-order}}$ and Equation 6, it is easy to notice that $\widehat{\nabla}_{\text{1st-order}}$ approximates $f(\boldsymbol{I}_i) - f(\boldsymbol{I}_j)$ as $\frac{\partial f(\boldsymbol{I}_j)}{\partial \boldsymbol{I}_j}(\boldsymbol{I}_i - \boldsymbol{I}_j)$. In numerical analyses, this approximation is known as the forward Euler method, which has first-order accuracy (we provide a brief introduction to the forward Euler method in Appendix E). Correspondingly, we know that $\widehat{\nabla}_{\text{1st-order}}$ is a first-order approximation of $\nabla$.

Now, we proceed to show $\widehat{\nabla}_{\text{ST}}$ works as a first-order approximation. Note that our analyses only apply to $\widehat{\nabla}_{\text{ST}}$ as defined in Equation 4 and may not apply to its other variants.

**Theorem 3.1.**

$$E[\widehat{\nabla}_{ST}] = \widehat{\nabla}_{1st\text{-}order}.$$

The proof of Theorem 3.1 is provided in Appendix A.

It is worth mentioning that Tokui & Sato (2017) discussed this connection for the special case of $\boldsymbol{D}$ being a Bernoulli variable. However, their study is built upon a Bernoulli variable property (i.e., $\nabla = (f(\boldsymbol{I}_2) - f(\boldsymbol{I}_1))\frac{d\boldsymbol{\pi}_1}{d\theta} = (f(\boldsymbol{I}_1) - f(\boldsymbol{I}_2))\frac{d\boldsymbol{\pi}_2}{d\theta}$), making their analyses not applicable to multinomial variables. Alternatively, the analyses in Gregor et al. (2014) and Pervez et al. (2020) are applicable to multinomial variables but resort to modify $\widehat{\nabla}_{\text{ST}}$ as $\frac{1}{n\cdot\boldsymbol{\pi}_D}\widehat{\nabla}_{\text{ST}}$, in order to position it as a first-order approximation. We suggest that this modification would lead to unwanted instability and provide more discussions in Section 4.1 and Section 6.6. Here, our study is the first to formally established $\widehat{\nabla}_{\text{ST}}$ works as a first-order approximation in the general multinomial case.

Besides revealing the mechanism of the Straight-Through estimator, our finding also shows that the bias of $\widehat{\nabla}_{\text{ST}}$ comes from using the first-order approximation (i.e., the forward Euler method). Accordingly, we propose to integrate a better approximation for $f(\boldsymbol{I}_i) - f(\boldsymbol{I}_j)$.

## 3.2 Towards Second-order Accuracy: ReinMax

The literature on numerical methods for differential equations shows that it is possible to achieve higher-order accuracy *without computing higher-order derivatives*. Correspondingly, we propose to integrate a second-order approximation to reduce the bias of the gradient estimator.

**Definition 3.2.** *One second-order approximation of $\nabla$ is*

$$\widehat{\nabla}_{2nd\text{-}order} := \sum_i \sum_j \frac{\boldsymbol{\pi}_j}{2}\left(\frac{\partial f(\boldsymbol{I}_j)}{\partial \boldsymbol{I}_j} + \frac{\partial f(\boldsymbol{I}_i)}{\partial \boldsymbol{I}_i}\right)(\boldsymbol{I}_i - \boldsymbol{I}_j)\frac{d\boldsymbol{\pi}_i}{d\boldsymbol{\theta}}.$$

Comparing $\widehat{\nabla}_{\text{2nd-order}}$ and Equation 6, we can observe that, $\widehat{\nabla}_{\text{2nd-order}}$ approximates $f(\boldsymbol{I}_i) - f(\boldsymbol{I}_j)$ as $\frac{1}{2}\left(\frac{\partial f(\boldsymbol{I}_i)}{\partial \boldsymbol{I}_i} + \frac{\partial f(\boldsymbol{I}_j)}{\partial \boldsymbol{I}_j}\right)(\boldsymbol{I}_i - \boldsymbol{I}_j)$. This approximation is known as the Heun's Method and has second-order accuracy (we provide a brief introduction to Heun's method in Appendix E). Correspondingly, we know that $\widehat{\nabla}_{\text{2nd-order}}$ is a second-order approximation of $\nabla$.

Based on this approximation, we propose the ReinMax operator as ($\boldsymbol{\pi}_D$ refers to $\frac{\boldsymbol{\pi}+\boldsymbol{D}}{2}$, $\mathsf{I}$ refers to the identity matrix, and $\odot$ refers to the element-wise product):

$$\widehat{\nabla}_{\text{ReinMax}} := 2 \cdot \widehat{\nabla}^{\frac{\boldsymbol{\pi}+\boldsymbol{D}}{2}} - \frac{1}{2}\widehat{\nabla}_{\text{ST}}, \text{ where } \widehat{\nabla}^{\frac{\boldsymbol{\pi}+\boldsymbol{D}}{2}} = \frac{\partial f(\boldsymbol{D})}{\partial \boldsymbol{D}} \cdot ((\boldsymbol{\pi}_D \cdot 1^T) \odot \mathsf{I} - \boldsymbol{\pi}_D \cdot \boldsymbol{\pi}_D^T) \quad (7)$$

---

[4]Please note that $\sum_i E[f(\boldsymbol{D})]\frac{d\boldsymbol{\pi}_i}{d\boldsymbol{\theta}} = E[f(\boldsymbol{D})]\frac{d\sum_i \boldsymbol{\pi}_i}{d\boldsymbol{\theta}} = E[f(\boldsymbol{D})]\frac{d\mathbf{1}}{d\boldsymbol{\theta}} = 0$

Then, we show that $\widehat{\nabla}_{\text{ReinMax}}$ approximates $\nabla$ to the second order. Or, formally we have:

**Theorem 3.2.**

$$E[\widehat{\nabla}_{ReinMax}] = \widehat{\nabla}_{2nd\text{-}order}.$$

The proof of Theorem 3.2 is provided in Appendix B.

**Computation Efficiency of ReinMax.** Instead of requiring Hessian or other second-order derivatives, $\widehat{\nabla}_{\text{ReinMax}}$ achieves second-order accuracy with two first-order derivatives (i.e., $\frac{\partial f(I_j)}{\partial I_j}$ and $\frac{\partial f(I_i)}{\partial I_i}$). As observed in our empirical efficiency comparisons in Section 6, the computation overhead of $\widehat{\nabla}_{\text{ReinMax}}$ is negligible. At the same time, similar to $\widehat{\nabla}_{\text{ST}}$ (as in Algorithm 1), our proposed algorithm can be easily integrated with existing automatic differentiation toolkits like PyTorch (a simple implementation of ReinMax is provided in Algorithm 2), making it easy to be integrated with existing algorithms.

**Applicability of Higher-order ODE solvers.** Although it's possible to apply higher-order ODE solvers, they require more gradient evaluations, leading to undesirable computational overhead. To illustrate this point: The approximation used by ReinMax requires n gradient evaluations, i.e., $\{\frac{\partial f(I_i)}{\partial I_i}\}$. In contrast, the approximation derived by RK4 needs $n^2 + n$ gradient evaluations, i.e., $\{\frac{\partial f(I_i)}{\partial I_i}\}$ and $\{\frac{\partial f(I_{ij})}{\partial I_{ij}}\}$, where $I_{ij} = \frac{I_i + I_j}{2}$. Therefore, while higher-order solvers are applicable, they may not be suitable in our case.

## 4 ReinMax and Baseline Subtraction

Equation 6 plays a crucial role in positioning ST as a first-order approximation of the gradient and deriving our proposed method, ReinMax. This equation is commonly referred to as baseline subtraction, a common technique for reducing the variance of REINFORCE.

In this section, we first discuss the reason for choosing $E[f(D)]$ as the baseline, and then reveal that the derivation of ReinMax is independent to baseline subtraction.

### 4.1 Benefits of Choosing $E[f(\mathbf{D})]$ as the Baseline

The choice of baseline in reinforcement learning has been the subject of numerous discussions (Weaver & Tao, 2001; Rennie et al., 2016; Shi et al., 2022). Similarly, in our study, different baselines lead to different gradient approximations.

Here, we discuss the rationale for choosing $E[f(D)]$ as the baseline. Considering $\sum_i \phi_i f(I_i)$ as the general form of the baseline ($\phi_i$ is a distribution over $\{I_1, \cdots, I_n\}$, i.e., $\sum_i \phi_i = 1$), we have:

**Remark 4.1.** *When $\sum_i \phi_i f(I_i)$ is used as the baseline and $f(I_i) - f(I_j)$ is approximated as $\frac{\partial f(I_j)}{\partial I_j}(I_i - I_j)$, we mark the resulting first-order approximation of $\nabla$ as $\widehat{\nabla}_{1st\text{-}order\text{-}avg\text{-}baseline}$. Then, we have $E[\frac{\phi_D}{\pi_D}\widehat{\nabla}_{ST}] = \widehat{\nabla}_{1st\text{-}order\text{-}avg\text{-}baseline}$.*

The derivations of Remark 4.1 are provided in Appendix C. Intuitively, since $\pi_D$ is the output of the softmax function, it could have very small values, which makes $\frac{\phi_D}{\pi_D}$ to be unreasonably large and leads to undesired instability. Therefore, we suggest that $E[f(D)]$ is a better choice of baseline when it comes to gradient approximation, since its corresponding gradient approximation is free of the instability $\frac{\phi_D}{\pi_D}$ brought.

It is worth mentioning that, when setting $\phi$ as $\frac{1}{n}$, the result of Remark 4.1 echoes some existing studies. Specifically, both Gregor et al. (2014) and Pervez et al. (2020) propose to approximate $\nabla$ as $E[\frac{1}{n \cdot \pi_D}\widehat{\nabla}_{ST}]$, which matches the result of Remark 4.1 by setting $\phi = \frac{1}{n}$.

In Section 6, we compared the corresponding second-order approximation when treating $E[f(D)]$ and $\frac{1}{n}\sum_i f(I_i)$ as the baseline, respectively. We observed that gradient estimators that use $E[f(D)]$ as the baseline consistently outperform gradient estimators that use $\frac{1}{n}\sum_i f(I_i)$ as the baseline, which verifies our intuition and demonstrates the importance of the baseline selection.

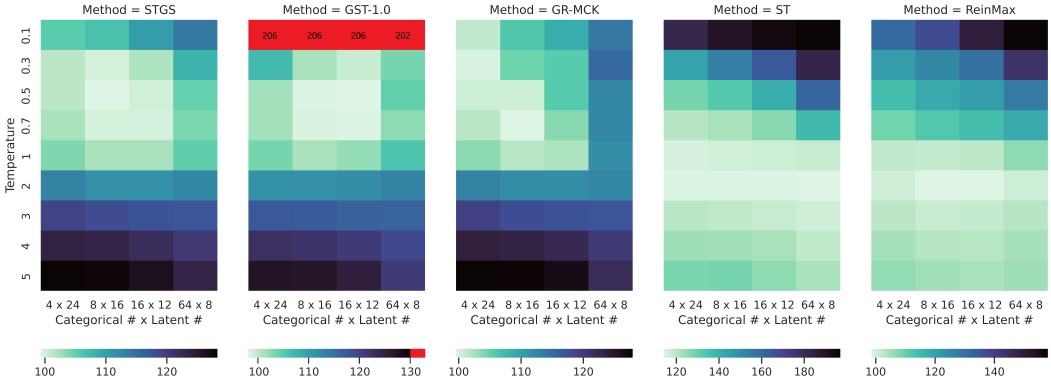

Figure 2: Training $-$ELBO on MNIST-VAE (lighter color indicates better performance). STGS, GST-1.0, and GR-MCK prefer to set the temperature $\tau \leq 1$. ST and ReinMax prefer to set $\tau \geq 1$.

## 4.2 Independence of ReinMax over Baseline Subtraction

To better understand the effectiveness of ReinMax, we further provide an alternative derivation that does not rely on the selection of the baseline. For simplicity, we only discuss $\frac{\partial \mathcal{L}}{\partial \boldsymbol{\theta}_k}$ and mark it as $\nabla_k$. Similar to Equation 2, we have:

$$\nabla_k := \frac{\partial \mathcal{L}}{\partial \boldsymbol{\theta}_k} = \sum_i f(\boldsymbol{I}_i) \frac{d \boldsymbol{\pi}_i}{d \boldsymbol{\theta}_k} = \boldsymbol{\pi}_k \sum_i \boldsymbol{\pi}_i (f(\boldsymbol{I}_k) - f(\boldsymbol{I}_i)). \tag{8}$$

It is worth mentioning that the derivation of Equation 8 leverages the derivative of the softmax function (i.e., for $\boldsymbol{\pi} = \mathrm{softmax}(\boldsymbol{\theta})$, we have $\partial \boldsymbol{\pi}_i / \partial \boldsymbol{\theta}_k = \boldsymbol{\pi}_k(\delta_{ik} - \boldsymbol{\pi}_i)$) and does not involve the baseline subtraction technology.

**Remark 4.2.** *In Equation 8, we approximate* $f(\boldsymbol{I}_k) - f(\boldsymbol{I}_i)$ *as* $\frac{1}{2}(\frac{\partial f(\boldsymbol{I}_i)}{\partial \boldsymbol{I}_i} + \frac{\partial f(\boldsymbol{I}_k)}{\partial \boldsymbol{I}_k})(\boldsymbol{I}_k - \boldsymbol{I}_i)$, *and mark the resulting second-order approximation of* $\nabla_k$ *as* $\widehat{\nabla}_{\textit{2nd-order-wo-baseline},k} = \boldsymbol{\pi}_k \sum_i \boldsymbol{\pi}_i \frac{1}{2}(\frac{\partial f(\boldsymbol{I}_i)}{\partial \boldsymbol{I}_i} + \frac{\partial f(\boldsymbol{I}_k)}{\partial \boldsymbol{I}_k})(\boldsymbol{I}_k - \boldsymbol{I}_i)$, *Then, we have* $E[\widehat{\nabla}_{\textit{ReinMax}}] = \widehat{\nabla}_{\textit{2nd-order-wo-baseline}}$

The proof of Remark 4.2 is provided in Appendix D.

As in Remark 4.2, applying the Heun's method on Equation 8 and Equation 6 lead to the same gradient estimator, which implies another benefit of using $E[f(\boldsymbol{D})]$ as the baseline: the resulting gradient estimator does not rely on additional prior (i.e., its derivation can be free of baseline subtraction).

## 5 Temperature Scaling for Gradient Estimators

Here, we discuss how to apply temperature scaling, a technique widely used in gradient estimators, to our proposed method, ReinMax. While the typical practice is to set the temperature $\tau$ to small values for STGS, we show that ST and ReinMax need a different strategy.

**Temperature Scaling for $\nabla_{\textbf{STGS}}$.** As introduced in Section 2, $\nabla_{\mathrm{STGS}}$ conduct a two-step approximation: (1) it approximates $\min_\theta E[f(\boldsymbol{D})]$ as $\min_\theta E[f(\mathrm{softmax}_\tau(\boldsymbol{\theta} + \boldsymbol{G})))]$; (2) it approximates $\frac{\partial f(\mathrm{softmax}_\tau(\boldsymbol{\theta}+\boldsymbol{G}))}{\partial \mathrm{softmax}_\tau(\boldsymbol{\theta}+\boldsymbol{G})}$ as $\frac{f(\boldsymbol{D})}{\partial \boldsymbol{D}}$. Since the bias introduced in both steps can be controlled by $\tau$, $\nabla_{\mathrm{STGS}}$ prefers to set $\tau$ as a relatively small value.

**Temperature Scaling for $\nabla_{\textbf{ST}}$ and $\nabla_{\textbf{ReinMax}}$.** As in Section 4, it does not involve temperature scaling to show $\nabla_{\mathrm{ST}}$ and $\nabla_{\mathrm{ReinMax}}$ work as the first-order and the second-order approximation to the gradient. Correspondingly, temperature scaling technology cannot help to reduce the bias for $\nabla_{\mathrm{ST}}$ in the same way it does for $\nabla_{\mathrm{STGS}}$. As in Figure 2, STGS, GR-MCK, and GST-1.0 work better when setting the temperature $\tau \leq 1$. ST and ReinMax work better when setting the temperature $\tau \geq 1$.

Thus, we incorporate temperature scaling to smooth the gradient approximation ($\boldsymbol{\pi}_\tau = \mathrm{softmax}_\tau(\boldsymbol{\theta})$) as $\widehat{\nabla}_{\mathrm{ReinMax}} = 2 \cdot \widehat{\nabla}^{\frac{\boldsymbol{\pi}_\tau + \boldsymbol{D}}{2}} - \frac{1}{2}\widehat{\nabla}_{\mathrm{ST}}$. It is worth emphasizing that $\tau$ in $\widehat{\nabla}_{\mathrm{ReinMax}}$ is used to stabilize the gradient approximation (instead of reducing bias) at the cost of accuracy. Therefore, the value of $\tau$ should be larger or equal to 1.

Table 1: Performance on ListOps.

| | STGS | GR-MCK | GST-1.0 | ST | ReinMax |
|---|---|---|---|---|---|
| Valid Accuracy | 66.95±3.05 | 66.53±0.58 | 66.28±0.52 | 66.51±0.76 | **67.65±1.25** |
| Test Accuracy | 67.30±2.50 | 66.53±0.86 | 66.30±0.62 | 66.26±0.48 | **68.07±1.18** |

Table 2: Training $-$ELBO on MNIST ($N \times M$ refers to $N$ categorical dim. and $M$ latent dim.).

| | AVG | $8 \times 4$ | $4 \times 24$ | $8 \times 16$ | $16 \times 12$ | $64 \times 8$ | $10 \times 30$ |
|---|---|---|---|---|---|---|---|
| STGS | 105.20 | 126.85±0.85 | 101.32±0.43 | 99.32±0.33 | 100.09±0.32 | 104±0.41 | 99.63±0.63 |
| GR-MCK | 107.06 | 125.94±0.71 | 99.96±0.25 | 99.58±0.31 | 102.54±0.48 | 112.34±0.48 | 102.02±0.18 |
| GST-1.0 | 104.25 | 126.35±1.24 | 101.49±0.44 | 98.29±0.66 | 98.12±0.57 | 102.53±0.57 | 98.64±0.33 |
| ST | 116.72 | 135.53±0.31 | 112.03±0.03 | 112.94±0.32 | 113.31±0.43 | 113.90±0.28 | 112.63±0.34 |
| ReinMax | **103.21** | **124.66±0.88** | **99.77±0.45** | **97.70±0.39** | **98.06±0.53** | **100.71±0.70** | **98.37±0.44** |

# 6 Experiments

Here, we conduct experiments on polynomial programming, unsupervised generative modeling, and structured output prediction. In all experiments, we consider four major baselines: Straight-Through (ST), Straight-Through Gumbel-Softmax (STGS), Gumbel-Rao Monte Carlo (GR-MCK), and Gapped Straight-Through (GST-1.0). For a more comprehensive comparison, we run a complete grid search on the training hyper-parameters for all methods. Also, we would reference results from the literature when their setting is comparable with ours. More details are elaborated in Appendix F.

## 6.1 Polynomial Programming

Following previous studies (Tucker et al., 2017; Grathwohl et al., 2018; Pervez et al., 2020; Paulus et al., 2021), we start with a simple problem. Consider $L$ i.i.d. latent binary variables $\boldsymbol{X}_1, \cdots, \boldsymbol{X}_L \in \{0, 1\}$ and a constant vector $\boldsymbol{c} \in \mathcal{R}^{L \times 1}$, we parameterize the distributions of $\{\boldsymbol{X}_1, \cdots, \boldsymbol{X}_L\}$ with $L$ softmax functions, i.e., $\boldsymbol{X}_i \overset{\text{iid}}{\sim} \text{Multinomial}(\text{softmax}(\boldsymbol{\theta}_i))$ and $\boldsymbol{\theta}_i \in \mathcal{R}^2$. Following previous studies, we set every dimension of $\boldsymbol{c}$ as $0.45$, i.e., $\forall i, \boldsymbol{c}_i = 0.45$, and use $\min_{\boldsymbol{\theta}} E_{\boldsymbol{X}}\left[\frac{\|\boldsymbol{X} - \boldsymbol{c}\|_p^p}{L}\right]$ as the objective.

**Training Curve with Various $p$.** We first set the number of latent variables (i.e., $L$) as 128 and batch size as 256. The training curve is visualized in Figure 1 for $p = 1.5$, 2, and 3. In all cases, ReinMax achieved near-optimal performance and the best convergence speed. Meanwhile, we can observe that ST and GST-1.0 do not perform well in all three cases. Although the final performance of STGS and GR-MCK is close to ReinMax, ReinMax has a faster convergence speed.

## 6.2 ListOps

We conducted unsupervised parsing on ListOps (Nangia & Bowman, 2018) and summarized the average accuracy and the standard derivation in Table 1. We also visualized the accuracy and loss on the valid set in Figure 3. Although the ST algorithm performs poorly on polynomial programming, it achieves a reasonable performance on this task. Also, while all baseline methods perform similarly, our proposed method stands out and brings consistent improvements. This further demonstrates the benefits of achieving second-order accuracy and the effectiveness of our proposed method.

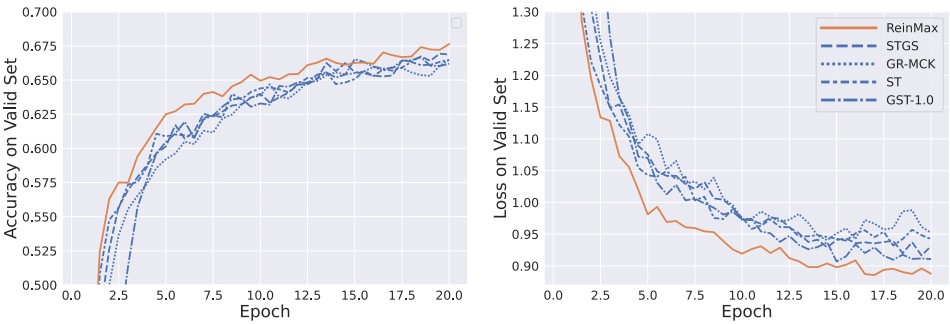

Figure 3: The accuracy (left) and loss (right) on the valid set of ListOps.

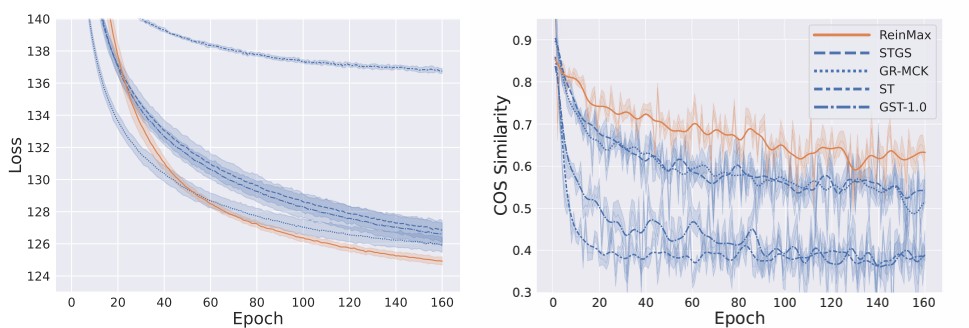

Figure 4: The training −ELBO (left) and the cos similarity between the gradient and its approximations (right) on MNIST-VAE (with 4 latent dimensions and 8 categorical dimensions).

Table 3: Average time cost (per epoch) / peak memory consumption on quadratic programming (QP) and MNIST-VAE. QP is configured to have 128 binary latent variables and 512 samples per batch. MNIST-VAE is configured to have 10 categorical dimensions and 30 latent dimensions.

|  | ReinMax | ST | STGS | GST-1.0 | GR-MCK$_{100}$ | GR-MCK$_{300}$ | GR-MCK$_{1000}$ |
|---|---|---|---|---|---|---|---|
| QP | 0.2s / 6.5Mb | 0.2s / 5.0Mb | 0.2s / 5.5Mb | 0.2s / 8.0Mb | 0.8s / 0.3Gb | 2.2s / 1Gb | 6.6s / 3Gb |
| MNIST-VAE | 5.2s / 13Mb | 5.2s / 13Mb | 5.2s / 13Mb | 5.2s / 13Mb | 5.2s / 76Mb | 5.2s / 0.2Gb | 5.4s / 0.6Gb |

Table 4: Performance on NATS-Bench. * Baseline results are referenced from Dong et al. (2020a).

|  | CIFAR-10 | | CIFAR-100 | | ImageNet-16-120 | |
|---|---|---|---|---|---|---|
|  | validation | test | validation | test | validation | test |
| GDAS + STGS* | 89.68±0.72 | 93.23±0.58 | 68.35±2.71 | 68.17±2.50 | 39.55±0.00 | 39.40±0.00 |
| GDAS + ReinMax | **90.01±0.12** | **93.44±0.23** | **69.29±2.34** | **69.41±2.24** | **41.47±0.79** | **42.03±0.41** |

## 6.3 MNIST-VAE

We benchmark the performance by training variational auto-encoders (VAE) with *categorical* latent variables on MNIST (LeCun et al., 1998). As we aim to compare gradient estimators, we focus our discussions on training ELBO. We find that training performance largely mirrors test performance (Dong et al., 2020b, 2021; Fan et al., 2022) and briefly discussed test ELBO in Appendix F.

**Biases of the Approximated Gradient.**    With 4 latent dimensions and 8 categorical dimensions, we iterate through the whole latent space (the size of the latent space is only 4096), compute the gradient as in Equation 2, and measured the cosine similarity between the gradient of latent variables and various approximations. As visualized in Figure 4, ReinMax achieves consistently more accurate gradient approximation across the training and, accordingly, faster convergence. Also, we can observe that, besides faster convergence, the performance of ReinMax is more stable.

**Experiment with Larger Latent Spaces.**    Let us proceed to larger latent spaces. First, we consider 4 settings with the latent space of $2^{48}$. Then, following Fan et al. (2022), we also conduct experiments with 10 latent dimensions and 30 categorical dimensions (the size of the latent space is $10^{30}$). As summarized in Table 2, ReinMax achieves the best performance on all configurations.

**GST-1.0 Performance on Different Problems.**    It is worth mentioning that, despite GST-1.0 achieving good performance on most settings of MNIST-VAE, it fails to maintain this performance on polynomial programming and unsupervised parsing, as discussed before. Upon discussing with Fan et al. (2022), we suggest that this phenomenon is caused by the characteristic of GST-1.0, which behaves similarly to ST on problems with a near one-hot optimal distribution. In other words, GST-1.0 has an implicit prior and prefers distributions that are not one-hot. At the same time, a different variant of GST (i.e., GST-p) would behave similarly to STGS on problems with a near one-hot optimal distribution, which achieves a significant performance boost over GST-1.0 on polynomial programming. However, on MNIST-VAE and ListOps, GST-p achieves an inferior performance.

This observation verifies our intuition that, without understanding the mechanism of ST, different applications have different preferences on its configurations. Meanwhile, ReinMax achieves consistent improvements in all settings, which greatly simplifies future algorithms developments.

Table 5: Training −ELBO on MNIST. * All baseline results are referenced from Fan et al. (2022)

|  | RLOO* | DisARM-Tree* | STGS* | GR-MCK* | GST-1.0* | ST* | ReinMax |
|---|---|---|---|---|---|---|---|
| Neg. ELBO | 104.03±0.23 | 103.10±0.25 | 97.32±0.20 | 110.74±1.23 | 96.09±0.25 | 116±0.09 | **93.44±0.51** |

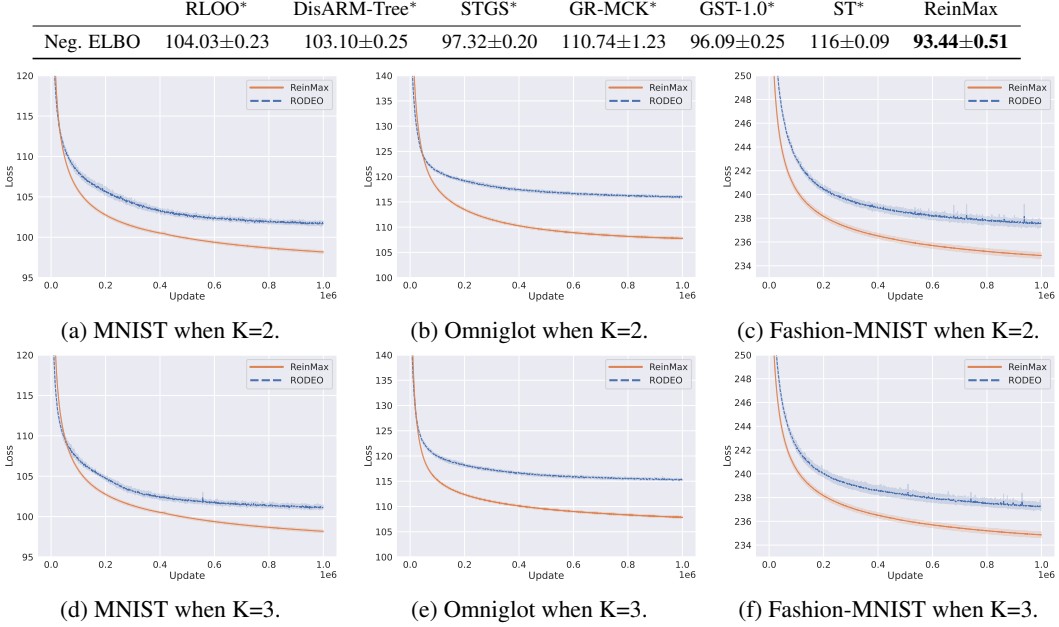

(a) MNIST when K=2.    (b) Omniglot when K=2.    (c) Fashion-MNIST when K=2.

(d) MNIST when K=3.    (e) Omniglot when K=3.    (f) Fashion-MNIST when K=3.

Figure 5: 2×200 VAE training curves on MNIST, Omniglot, and Fashion-MNIST when K=2 or 3.

Table 6: Train −ELBO of $2 \times 200$ VAE on MNIST, Fashion-MNIST, and Omniglot. * Baseline results are referenced from Shi et al. (2022). K refers to the number of evaluations.

|  |  | RELAX* | ARMS* | DisARM* | Double CV* | RODEO* | ReinMax |
|---|---|---|---|---|---|---|---|
| K=3 | MNIST | 101.99±0.04 | 100.84±0.14 | / | 100.94±0.09 | 100.46±0.13 | **97.83±0.36** |
|  | Fashion-MNIST | 237.74±0.12 | 237.05±0.12 | / | 237.40±0.11 | 236.88±0.12 | **234.53±0.42** |
|  | Omniglot | 115.70±0.08 | 115.32±0.07 | / | 115.06±0.12 | 115.01±0.05 | **107.51±0.42** |
| K=2 | MNIST | / | / | 102.75±0.08 | 102.14±0.06 | 101.89±0.17 | **98.17±0.29** |
|  | Fashion-MNIST | / | / | 237.68±0.13 | 237.55±0.16 | 237.44±0.09 | **234.89±0.21** |
|  | Omniglot | / | / | 116.50±0.04 | 116.39±0.10 | 115.93±0.06 | **107.79±0.27** |

## 6.4 Applying ReinMax to Differentiable Neural Architecture Search

To demonstrate the applicability of ReinMax as a drop-in replacement, we conduct experiments following the topology search setting in the NATS-Bench benchmark (Dong et al., 2020a), and summarize the results in Table 4. GDAS is an algorithm that employs STGS to estimate the gradient of neural architecture parameters (Dong & Yang, 2019). We replaced STGS with ReinMax as the gradient estimator (configurations elaborated in Appendix F). ReinMax brings consistent performance improvements across all three datasets, demonstrating the great potential of ReinMax.

## 6.5 Comparisons with REINFORCE-style Methods

Here, we conduct experiments to discuss the difference between ReinMax and REINFORCE-style methods. First, following Fan et al. (2022), we conduct experiments on the setting with a larger batch size (i.e., 200), longer training (i.e., $5 \times 10^5$ steps), 32 latent dimensions, and 64 categorical dimensions (details are elaborated in Appendix F). As in Table 5, ReinMax outperforms all baselines, including two REINFORCE-based methods (Dong et al., 2020b, 2021).

We further conduct experiments to compare with the state of the art. Specifically we apply ReinMax to Bernoulli VAEs on MNIST, Fashion-MNIST (Xiao et al., 2017), and Omniglot(Lake et al., 2015), adhering closely to the experimental settings of Shi et al. (2022), including pre-processing, model architecture, batch size, and training epochs. As in Tables 6 and Figure 5, ReinMax consistently outperforms RODEO across all settings. To better understand the difference between RODEO and ReinMax, we conduct more experiments on polynomial programming (as elaborated in Appendix F.6).

Overall, ReinMax achieves better performance in more challenging scenarios, i.e., smaller batch size, more latent variables, or more complicated problems. Meanwhile, REINFORCE and RODEO achieve better performance on simpler problem settings, i.e., larger batch size, fewer latent variables, or simpler problems. This observation matches our intuition:

- REIFORCE-style algorithms excel as they provide unbiased gradient estimation but may fall short in complex scenarios, since they only utilize the zero-order information (i.e., a scalar $f(\boldsymbol{D})$).

- ReinMax, using more information (i.e., a vector $\frac{\partial f(\boldsymbol{D})}{\partial \boldsymbol{D}}$), handles challenging scenarios better. Meanwhile, as a consequence of its estimation bias, ReinMax leads to slower convergence in some simple scenarios.

### 6.6 Discussions

**Choice of Baseline.** As introduced in Section 4.1, the choice of subtraction baseline has a huge impact on the performance. Here, we demonstrate this empirically.

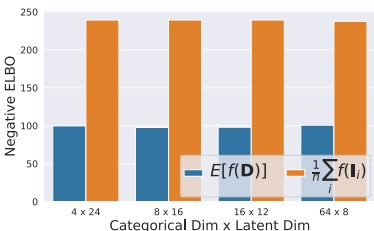

We use $\frac{1}{n} \sum_i f(\boldsymbol{I}_i)$ as the baseline and compare the resulting gradient approximation with ReinMax. As visualized in Figure 6, ReinMax, which uses $E[f(\boldsymbol{D})]$ as the baseline, significantly outperforms the one that uses $\frac{1}{n} \sum_i f(\boldsymbol{I}_i)$ as the baseline. We suspect that the gradient approximation using $\frac{1}{n} \sum_i f(\boldsymbol{I}_i)$ as the baseline is very unstable as it contains the $\frac{1}{n \cdot p(\boldsymbol{D})}$ term.

Figure 6: Training $-$ELBO on MNIST-VAE when using $\frac{1}{n} \sum_i f(\boldsymbol{I}_i)$ and $E[f(\boldsymbol{D})]$ as baselines respectively.

**Temperature Scaling.** On MNIST-VAE (four settings with the $2^{48}$ latent space), we utilize heatmaps to visualize the final performance of all five methods under different temperatures, i.e., $\{0.1, 0.3, 0.5, 0.7, 1, 2, 3, 4, 5\}$. As in Figure 2, these methods have different preferences for the temperature configuration. Specifically, STGS, GST-1.0, and GR-MCK prefer to set the temperature $\tau \leq 1$. Differently, ST and ReinMax prefer to set the temperature $\tau \geq 1$. These observations match our analyses in Section 5 that a small $\tau$ can help reduce the bias introduced by STGS-style methods. Also, it verifies that ST and ReinMax work differently from STGS, GST-1.0, and GR-MCK.

**Efficiency.** As summarized in Table 3, we can observe that, since GR-MCK uses the Monte Carlo method to reduce the variance, it has larger time and memory consumption, which becomes less significant with fewer Monte Carlo samples (we use GR-MCK$_s$ to indicate GR-MCK with $s$ Monte Carlo samples). Meanwhile, all remaining methods have roughly the same time and memory consumption. This shows that the computation overheads of ReinMax are negligible.

## 7 Conclusion and Future Work

In this study, we seek the underlying principle of the Straight-Through (ST) gradient estimator. We formally establish that ST works as a first-order approximation of the gradient and propose a novel method, ReinMax, which incorporates Heun's Method and achieves second-order accuracy without requiring second-order derivatives. We conduct extensive experiments on polynomial programming, unsupervised generative modeling, and structured output prediction. ReinMax brings consistent improvements over the state-of-the-art methods.

It is worth mentioning that analyses in this study further guided us to empower Mixture-of-Expert training (Liu et al., 2023). Specifically, for gradient approximation of sparse expert routing, while ReinMax requires the network to be fully activated, Liu et al. (2023) uses $f(\boldsymbol{0})$ as the baseline and only requires the network to be partially activated. In the future, we plan to conduct further analyses on the truncation error to stabilize and improve the gradient estimation.

## Acknowledgement

We would like to thank all reviewers for their constructive comments, the engineering team at Microsoft for providing computation infrastructure support, Alessandro Sordoni, Nicolas Le Roux, and Greg Yang for their helpful discussions.

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

## A  Theorem 3.1

Let us define the first-order approximation of $\nabla$ as $\widehat{\nabla}_{\text{1st-order}} = \sum_i \sum_j \boldsymbol{\pi}_j \frac{\partial f(\boldsymbol{I}_j)}{\partial \boldsymbol{I}_j}(\boldsymbol{I}_i - \boldsymbol{I}_j)\frac{d\boldsymbol{\pi}_i}{d\boldsymbol{\theta}}$, which approximates $f(\boldsymbol{I}_i) - f(\boldsymbol{I}_j)$ in Equation 6 as $\frac{\partial f(\boldsymbol{I}_j)}{\partial \boldsymbol{I}_j}(\boldsymbol{I}_i - \boldsymbol{I}_j)$.

**Theorem 3.1.**

$$E[\widehat{\nabla}_{ST}] = \widehat{\nabla}_{\textit{1st-order}}.$$

*Proof.* Based on the definition, we have:

$$
\begin{aligned}
\widehat{\nabla}_{\text{1st-order}} &= \sum_i \sum_j \boldsymbol{\pi}_j \frac{\partial f(\boldsymbol{I}_j)}{\partial \boldsymbol{I}_j}(\boldsymbol{I}_i - \boldsymbol{I}_j)\frac{d\boldsymbol{\pi}_i}{d\boldsymbol{\theta}} \\
&= \sum_j \boldsymbol{\pi}_j \frac{\partial f(\boldsymbol{I}_j)}{\partial \boldsymbol{I}_j}\sum_i \boldsymbol{I}_i \frac{d\boldsymbol{\pi}_i}{d\boldsymbol{\theta}} - \sum_j \boldsymbol{\pi}_j \frac{\partial f(\boldsymbol{I}_j)}{\partial \boldsymbol{I}_j}\boldsymbol{I}_j \sum_i \frac{d\boldsymbol{\pi}_i}{d\boldsymbol{\theta}} \qquad (9)
\end{aligned}
$$

Since $\sum_i \boldsymbol{\pi}_i = 1$, we have $\sum_i \frac{d\boldsymbol{\pi}_i}{d\boldsymbol{\theta}} = 0$. Also, since $\boldsymbol{\pi} = \sum_i \boldsymbol{\pi}_i \boldsymbol{I}_i$, we have $\frac{d\boldsymbol{\pi}}{d\boldsymbol{\theta}} = \sum_i \boldsymbol{I}_i \frac{d\boldsymbol{\pi}_i}{d\boldsymbol{\theta}}$. Thus, together with Equation 9, we have:

$$
\begin{aligned}
\widehat{\nabla}_{\text{1st-order}} &= \sum_j \boldsymbol{\pi}_j \frac{\partial f(\boldsymbol{I}_j)}{\partial \boldsymbol{I}_j}\sum_i \boldsymbol{I}_i \frac{d\boldsymbol{\pi}_i}{d\boldsymbol{\theta}} \\
&= E[\frac{\partial f(\boldsymbol{D})}{\partial \boldsymbol{D}}\frac{d\boldsymbol{\pi}}{d\boldsymbol{\theta}}] = E[\widehat{\nabla}_{\text{ST}}].
\end{aligned}
$$

$\square$

## B  Theorem 3.2

**Theorem 3.2.**

$$E[\widehat{\nabla}_{\textit{ReinMax}}] = \widehat{\nabla}_{\textit{2nd-order}}.$$

*Proof.* Here, we aim to proof, $\forall k \in [1, n]$, we have $E[\widehat{\nabla}_{\text{ReinMax},k}] = \widehat{\nabla}_{\text{2nd-order},k}$. As defined in Equation 8, we have (note that $\delta_{\boldsymbol{DI}_k}$ is the indicator function of the event $\boldsymbol{D} = \boldsymbol{I}_k$):

$$
\begin{aligned}
\widehat{\nabla}_{\text{2nd-order},k} &= \sum_i \sum_j \frac{\boldsymbol{\pi}_j}{2}(\frac{\partial f(\boldsymbol{I}_j)}{\partial \boldsymbol{I}_j} + \frac{\partial f(\boldsymbol{I}_i)}{\partial \boldsymbol{I}_i})(\boldsymbol{I}_i - \boldsymbol{I}_j)\frac{d\boldsymbol{\pi}_i}{d\boldsymbol{\theta}_k} \\
&= \sum_i \sum_j \frac{\boldsymbol{\pi}_j \boldsymbol{\pi}_i (\delta_{ik} - \boldsymbol{\pi}_k)}{2}(\frac{\partial f(\boldsymbol{I}_j)}{\partial \boldsymbol{I}_j} + \frac{\partial f(\boldsymbol{I}_i)}{\partial \boldsymbol{I}_i})(\boldsymbol{I}_i - \boldsymbol{I}_j) \\
&= \sum_j \frac{\boldsymbol{\pi}_j \boldsymbol{\pi}_k}{2}(\frac{\partial f(\boldsymbol{I}_j)}{\partial \boldsymbol{I}_j} + \frac{\partial f(\boldsymbol{I}_k)}{\partial \boldsymbol{I}_k})(\boldsymbol{I}_k - \boldsymbol{I}_j) \\
&= \frac{\boldsymbol{\pi}_k}{2}\frac{\partial f(\boldsymbol{I}_k)}{\partial \boldsymbol{I}_k}(\boldsymbol{I}_k - \sum_j \boldsymbol{\pi}_j \boldsymbol{I}_j) + \sum_j \frac{\boldsymbol{\pi}_j \boldsymbol{\pi}_k}{2}\frac{\partial f(\boldsymbol{I}_j)}{\partial \boldsymbol{I}_j}(\boldsymbol{I}_k - \boldsymbol{I}_j) \\
&= \frac{1}{2}E_{\boldsymbol{D}\sim\boldsymbol{\pi}}[\delta_{\boldsymbol{DI}_k}\frac{\partial f(\boldsymbol{D})}{\partial \boldsymbol{D}}(\boldsymbol{I}_{\boldsymbol{D}} - \sum_j \boldsymbol{\pi}_j \boldsymbol{I}_j)] + \frac{1}{2}E_{\boldsymbol{D}\sim\boldsymbol{\pi}}[\boldsymbol{\pi}_k\frac{\partial f(\boldsymbol{D})}{\partial \boldsymbol{D}}(\boldsymbol{I}_k - \boldsymbol{I}_{\boldsymbol{D}})] \\
&= \frac{1}{2}E_{\boldsymbol{D}\sim\boldsymbol{\pi}}[\frac{\partial f(\boldsymbol{D})}{\partial \boldsymbol{D}}(\boldsymbol{\pi}_k(\boldsymbol{I}_k - \boldsymbol{I}_{\boldsymbol{D}}) + \delta_{\boldsymbol{DI}_k}(\boldsymbol{I}_{\boldsymbol{D}} - \sum_i \boldsymbol{\pi}_i \boldsymbol{I}_i))] \qquad (10)
\end{aligned}
$$

At the same time, based on the definition of $\widehat{\nabla}_{\text{ReinMax}}$, we have:

$$
\begin{aligned}
E[\widehat{\nabla}_{\text{ReinMax},k}] &= E_{\boldsymbol{D}\sim\boldsymbol{\pi}}[\frac{\partial f(\boldsymbol{D})}{\partial \boldsymbol{D}}(2 \cdot \frac{\boldsymbol{\pi}_k + \delta_{\boldsymbol{DI}_k}}{2}(\boldsymbol{D}_k - \sum_i \frac{\boldsymbol{\pi}_i + \delta_{\boldsymbol{DI}_k}}{2}\boldsymbol{I}_i) - \frac{\boldsymbol{\pi}_k}{2}(\boldsymbol{D}_k - \sum_i \boldsymbol{\pi}_i \boldsymbol{I}_i))] \\
&= \frac{1}{2}E_{\boldsymbol{D}\sim\boldsymbol{\pi}}[\frac{\partial f(\boldsymbol{D})}{\partial \boldsymbol{D}}(\boldsymbol{\pi}_k(\boldsymbol{I}_k - \boldsymbol{I}_{\boldsymbol{D}}) + \delta_{\boldsymbol{DI}_k}(\boldsymbol{I}_k - \sum_i \boldsymbol{\pi}_i \boldsymbol{I}_i))] \qquad (11)
\end{aligned}
$$

Since $\delta_{\boldsymbol{DI}_k}(\boldsymbol{I}_k - \sum_i \boldsymbol{\pi}_i \boldsymbol{I}_i) = \delta_{\boldsymbol{DI}_k}(\boldsymbol{I_D} - \sum_i \boldsymbol{\pi}_i \boldsymbol{I}_i)$, together with Equation 10 and 11, we have:

$$E[\widehat{\nabla}_{\text{ReinMax},k}] = \widehat{\nabla}_{\text{2nd-order},k}$$

$\square$

## C  Remark 4.1

**Remark 3.1.** *When $\sum_i \phi_i f(\boldsymbol{I}_i)$ is used as the baseline and $f(\boldsymbol{I}_i) - f(\boldsymbol{I}_j)$ is approximated as $\frac{\partial f(\boldsymbol{I}_j)}{\partial \boldsymbol{I}_j}(\boldsymbol{I}_i - \boldsymbol{I}_j)$, we mark the resulting first-order approximation of $\nabla$ as $\widehat{\nabla}_{1st\text{-}order\text{-}avg\text{-}baseline}$. Then, we have:*

$$E[\frac{\boldsymbol{\phi_D}}{\boldsymbol{\pi_D}}\widehat{\nabla}_{ST}] = \widehat{\nabla}_{1st\text{-}order\text{-}avg\text{-}baseline}$$

*Proof.* Using $\sum_i \phi_i f(\boldsymbol{I}_i)$ as the baseline, we have:

$$\nabla = \sum_i (f(\boldsymbol{I}_i) - \sum_j \phi_j f(\boldsymbol{I}_j))\frac{d\boldsymbol{\pi}_i}{d\boldsymbol{\theta}} = \sum_i \sum_j \phi_j(f(\boldsymbol{I}_i) - f(\boldsymbol{I}_j))\frac{d\boldsymbol{\pi}_i}{d\boldsymbol{\theta}}$$

Approximating $f(\boldsymbol{I}_i) - f(\boldsymbol{I}_j)$ as $\frac{\partial f(\boldsymbol{I}_j)}{\partial \boldsymbol{I}_j}(\boldsymbol{I}_i - \boldsymbol{I}_j)$, we have:

$$
\begin{aligned}
\widehat{\nabla}_{\text{1st-order-avg-baseline}} &= \sum_i \sum_j \phi_j \frac{\partial f(\boldsymbol{I}_j)}{\partial \boldsymbol{I}_j}(\boldsymbol{I}_i - \boldsymbol{I}_j)\frac{d\boldsymbol{\pi}_i}{d\boldsymbol{\theta}} \\
&= \sum_j \frac{\phi_j}{\boldsymbol{\pi}_j} \cdot \boldsymbol{\pi}_j \cdot \frac{\partial f(\boldsymbol{I}_j)}{\partial \boldsymbol{I}_j} \sum_i \boldsymbol{I}_i \frac{d\boldsymbol{\pi}_i}{d\boldsymbol{\theta}} \\
&= E[\frac{\boldsymbol{\phi_D}}{\boldsymbol{\pi_D}}\widehat{\nabla}_{\text{ST}}]
\end{aligned}
$$

$\square$

## D  Remark 4.2

**Remark 3.2.** *In Equation 8, we approximate $f(\boldsymbol{I}_k) - f(\boldsymbol{I}_i)$ as $\frac{1}{2}(\frac{\partial f(\boldsymbol{I}_i)}{\partial \boldsymbol{I}_i} + \frac{\partial f(\boldsymbol{I}_k)}{\partial \boldsymbol{I}_k})(\boldsymbol{I}_k - \boldsymbol{I}_i)$, and mark the resulting second-order approximation of $\nabla_k$ as $\widehat{\nabla}_{2nd\text{-}order\text{-}wo\text{-}baseline,k} = \boldsymbol{\pi}_k \sum_i \boldsymbol{\pi}_i \frac{1}{2}(\frac{\partial f(\boldsymbol{I}_i)}{\partial \boldsymbol{I}_i} + \frac{\partial f(\boldsymbol{I}_k)}{\partial \boldsymbol{I}_k})(\boldsymbol{I}_k - \boldsymbol{I}_i)$, Then, we have:*

$$E[\widehat{\nabla}_{ReinMax}] = \widehat{\nabla}_{2nd\text{-}order\text{-}wo\text{-}baseline}$$

*Proof.* Here, we aim to proof, $\forall k \in [1, n]$, we have $E[\widehat{\nabla}_{\text{ReinMax},k}] = \widehat{\nabla}_{\text{2nd-order-wo-baseline},k}$.

$$
\begin{aligned}
\widehat{\nabla}_{\text{2nd-order-wo-baseline},k} &= \boldsymbol{\pi}_k \sum_i \boldsymbol{\pi}_i \frac{1}{2}(\frac{\partial f(\boldsymbol{I}_i)}{\partial \boldsymbol{I}_i} + \frac{\partial f(\boldsymbol{I}_k)}{\partial \boldsymbol{I}_k})(\boldsymbol{I}_k - \boldsymbol{I}_i) \\
&= \boldsymbol{\pi}_k \sum_i \boldsymbol{\pi}_i \frac{1}{2}\frac{\partial f(\boldsymbol{I}_i)}{\partial \boldsymbol{I}_i}(\boldsymbol{I}_k - \boldsymbol{I}_i) + \boldsymbol{\pi}_k \sum_i \boldsymbol{\pi}_i \frac{1}{2}\frac{\partial f(\boldsymbol{I}_k)}{\partial \boldsymbol{I}_k}(\boldsymbol{I}_k - \boldsymbol{I}_i) \\
&= E[\frac{\partial f(\boldsymbol{D})}{\partial \boldsymbol{D}}\frac{\boldsymbol{\pi}_k(\boldsymbol{I}_k - \boldsymbol{I_D}) + \delta_{\boldsymbol{DI}_k}(\boldsymbol{I}_k - \sum_i \boldsymbol{\pi}_i \boldsymbol{I}_i)}{2}] = E[\widehat{\nabla}_{\text{ReinMax},k}]
\end{aligned}
$$

$\square$

# E  Forward Euler Method and Heun's Method

For simplicity, we consider a simple function $g(x) : \mathcal{R} \to \mathcal{R}$ that is three times differentiable on $[t_0, t_1]$. Now, we proceed to a simple introduction to approximate $\int_{t_0}^{t_1} g'(x)dx$ with the Forward Euler Method and the Heun's Method. For a detailed introduction to numerical ODE methods, please refer to Ascher & Petzold (1998).

**Forward Euler Method.** Here, we approximate $g(t_1)$ with the first-order Taylor expansion, i.e., $g(t_1) = g(t_0) + g'(t_0) \cdot (t_1 - t_0) + O((t_1 - t_0)^2)$, then we have $\int_{t_0}^{t_1} g'(x)dx \approx g'(t_0)(t_1 - t_0)$. Since we used the first-order Taylor expansion, this approximation has first-order accuracy.

**Heun's Method.** First, we approximate $g(t_1)$ with the second-order Taylor expansion:

$$g(t_1) = g(t_0) + g'(t_0) \cdot (t_1 - t_0) + \frac{g''(t_0)}{2} \cdot (t_1 - t_0)^2 + O((t_1 - t_0)^3). \tag{12}$$

Then, we show that we can match this approximation by combining the first-order derivatives of two samples. Taylor expanding $g'(t_1)$ to the first-order, we have:

$$g'(t_1) = g'(t_0) + g''(t_0) \cdot (t_1 - t_0) + O((t_1 - t_0)^2)$$

Therefore, we have:

$$g(t_0) + \frac{g'(t_0) + g'(t_1)}{2}(t_1 - t_0) = g(t_0) + g'(t_0) \cdot (t_1 - t_0) + \frac{g''(t_0)}{2} \cdot (t_1 - t_0)^2 + O((t_1 - t_0)^3).$$

It is easy to notice that the right-hand side of the above equation matches the second-order Taylor expansion of $g(t_1)$ as in Equation 12. Therefore, the above approximation (i.e., approximating $g(t_1) - g(t_0)$ as $\frac{g'(t_0)+g'(t_1)}{2}(t_1 - t_0)$) has second-order accuracy.

**Connection to $f(\boldsymbol{I}_i) - f(\boldsymbol{I}_j)$ in Equation 6.** By setting $g(x) = f(x \cdot \boldsymbol{I}_i + (1 - x) \cdot \boldsymbol{I}_j)$, we have $g(1) - g(0) = f(\boldsymbol{I}_i) - f(\boldsymbol{I}_j)$. Then, it is easy to notice that the forward Euler Method approximates $f(\boldsymbol{I}_i) - f(\boldsymbol{I}_j)$ as $\frac{\partial f(\boldsymbol{I}_j)}{\partial \boldsymbol{I}_j}(\boldsymbol{I}_i - \boldsymbol{I}_j)$ and has first-order accuracy. Also, the Heun's Method approximates $f(\boldsymbol{I}_i) - f(\boldsymbol{I}_j)$ as $\frac{1}{2}(\frac{\partial f(\boldsymbol{I}_i)}{\partial \boldsymbol{I}_i} + \frac{\partial f(\boldsymbol{I}_j)}{\partial \boldsymbol{I}_j})(\boldsymbol{I}_i - \boldsymbol{I}_j)$ and has second-order accuracy.

# F  Experiment Details

## F.1  Baselines

Here, we consider four methods as our major baselines:
- Straight-Through (ST; Bengio et al., 2013) backpropagate through the sampling function as if it had been the identity function.
- Straight-Through Gumbel-Softmax (STGS; Jang et al., 2017) integrates the Gumbel reparameterization trick to approximate the gradient.
- Gumbel-Rao Monte Carlo (GR-MCK; Paulus et al., 2021) leverages the Monte Carlo method to reduce the variance introduced by the Gumbel noise in STGS. To obtain the optimal performance for this baseline, we set the number of Monte Carlo samples to 1000 in most experiments. Except in our discussions of efficiency, we set the number of Monte Carlo samples to 100, 300, and 1000 for a more comprehensive comparisons.
- Gapped Straight-Through (GST-1.0; Fan et al., 2022) aims to reduce the variance of STGS and constructs a deterministic term to replace the Monte Carlo samples used in GR-MCK. Here, as suggested in (Fan et al., 2022), we set the gap (a hyper-parameter) as 1.0.

**GST-1.0 Performance.** Despite GST-1.0 achieving good performance on most settings of MNIST-VAE, it fails to maintain this performance on polynomial programming and unsupervised parsing, as discussed before. At the same time, a different variant of GST (i.e., GST-p) achieves a significant performance boost over GST-1.0 on polynomial programming. However, on MNIST-VAE and ListOps, GST-p achieves an inferior performance. Upon discussing with the author of the GST-1.0, we suggest that this phenomenon is caused by different characteristics of GST-1.0 and GST-p.

This observation verifies our intuition that, without understanding the mechanism of ST, different applications have different preferences on its configurations. Meanwhile, ReinMax achieves consistent improvements in all settings, which greatly simplifies future algorithms developments.

## F.2 Hyper-Parameters

Without specifically, we conduct full grid search for all methods in all experiments, and report the best performance (averaged with 10 random seeds on MNIST-VAE and 5 random seeds on ListOps). The hyper-parameter search space is summarized in Table 7. The search results for Table 2 and Table 1 are summarized in Table 8.

Table 7: Hyper-parameter search space.

| Hyperparameters | Search Space |
|---|---|
| Optimizer | {Adam(Kingma & Ba, 2015), RAdam(Liu et al., 2020)} |
| Learning Rate | {0.001, 0.0007, 0.0005, 0.0003} |
| Temperature | {0.1, 0.3, 0.5, 0.7, 1.0, 1.1, 1.2, 1.3, 1.4, 1.5} |

Table 8: Hyper-parameters Search Result for Results in Table 1 and Table 2.

| | | STGS | GR-MCK | GST-1.0 | ST | ReinMax |
|---|---|---|---|---|---|---|
| | Optimizer | Adam | Adam | Adam | Adam | Adam |
| MNIST-VAE $8 \times 4$ | Learning Rate | 0.0003 | 0.0005 | 0.0005 | 0.001 | 0.0005 |
| | Temperature | 0.5 | 0.5 | 0.7 | 1.3 | 1.3 |
| | Optimizer | RAdam | RAdam | RAdam | RAdam | RAdam |
| MNIST-VAE $4 \times 24$ | Learning Rate | 0.0005 | 0.0005 | 0.0005 | 0.001 | 0.0005 |
| | Temperature | 0.3 | 0.3 | 0.5 | 1.5 | 1.5 |
| | Optimizer | RAdam | RAdam | RAdam | RAdam | RAdam |
| MNIST-VAE $8 \times 16$ | Learning Rate | 0.0005 | 0.0007 | 0.0007 | 0.001 | 0.0007 |
| | Temperature | 0.5 | 0.7 | 0.5 | 1.5 | 1.5 |
| | Optimizer | RAdam | Adam | RAdam | Adam | RAdam |
| MNIST-VAE $16 \times 12$ | Learning Rate | 0.0007 | 0.0005 | 0.0007 | 0.0005 | 0.0007 |
| | Temperature | 0.7 | 1.0 | 0.5 | 1.5 | 1.5 |
| | Optimizer | RAdam | Adam | RAdam | Adam | RAdam |
| MNIST-VAE $64 \times 8$ | Learning Rate | 0.0007 | 0.0007 | 0.0007 | 0.0005 | 0.0005 |
| | Temperature | 0.7 | 2.0 | 0.7 | 1.5 | 1.5 |
| | Optimizer | RAdam | RAdam | RAdam | RAdam | RAdam |
| MNIST-VAE $10 \times 30$ | Learning Rate | 0.0005 | 0.0005 | 0.0005 | 0.0007 | 0.0005 |
| | Temperature | 0.5 | 1.0 | 0.5 | 1.4 | 1.3 |
| | Optimizer | RAdam | RAdam | RAdam | RAdam | RAdam |
| ListOps | Learning Rate | 0.0005 | 0.0005 | 0.001 | 0.001 | 0.0007 |
| | Temperature | 0.1 | 0.3 | 0.1 | 1.4 | 1.1 |

**Polynomial Programming.** As this problem is relatively simple, we set the learning rate to 0.001 and the optimizer to Adam, and only tune the temperature hyper-parameter.

**MNIST-VAE.** Following the previous study (Dong et al., 2020b, 2021; Fan et al., 2022), we used 2-layer MLP as the encoder and the decoder. We set the hidden state dimension of the first-layer and the second-layer as 512 and 256 for the encoder, and 256 and 512 for the decoder. For our experiments on MNIST-VAE with 32 latent dimensions and 64 categorical dimensions, we set the batch size to 200, training steps to $5 \times 10^5$, and activation function to LeakyReLU, in order to be consistent with the literature. For other experiments, we set the batch size to 100, the activation function to ReLU, and training steps to $9.6 \times 10^4$ (i.e., 160 epochs).

Table 9: Test −ELBO on MNIST. Hyper-parameters are chosen based on Train −ELBO.

|  | AVG | $8 \times 4$ | $4 \times 24$ | $8 \times 16$ | $16 \times 12$ | $64 \times 8$ | $10 \times 30$ |
|---|---|---|---|---|---|---|---|
| STGS | 106.89 | 128.09±0.79 | 103.60±0.45 | 99.32±0.33 | 102.49±0.32 | 106.20±0.46 | 101.61±0.54 |
| GR-MCK | 109.03 | 127.90±0.71 | 102.76±0.33 | 102.12±0.29 | 104.23±0.65 | 113.54±0.50 | 103.62±0.13 |
| GST-1.0 | 106.85 | 128.20±1.12 | 103.95±0.49 | 101.44±0.32 | 101.28±0.59 | 105.44±0.62 | 100.78±0.44 |
| ST | 118.85 | 137.06±0.51 | 113.41±0.49 | 114.25±0.29 | 114.48±0.56 | 115.43±0.29 | 118.46±0.18 |
| ReinMax | **105.74** | **126.89±0.79** | **102.40±0.43** | **100.63±0.41** | **100.85±0.50** | **102.91±0.67** | **100.75±0.50** |

Table 10: Test −ELBO on MNIST. Hyper-parameters are chosen based on Test −ELBO.

|  | AVG | $8 \times 4$ | $4 \times 24$ | $8 \times 16$ | $16 \times 12$ | $64 \times 8$ | $10 \times 30$ |
|---|---|---|---|---|---|---|---|
| STGS | 107.15 | 128.09±0.79 | 103.25±0.22 | 101.44±0.32 | 102.29±0.39 | 106.20±0.46 | 101.61±0.54 |
| GR-MCK | 108.87 | 127.86±0.54 | 102.40±0.37 | 101.59±0.22 | 104.22±0.63 | 113.54±0.50 | 103.62±0.13 |
| GST-1.0 | 106.55 | 128.03±1.02 | 103.63±0.24 | 100.67±0.34 | 101.04±0.39 | 105.44±0.62 | **100.51±0.37** |
| ST | 118.79 | 137.05±0.36 | 113.23±0.43 | 114.11±0.31 | 114.48±0.56 | 115.43±0.29 | 118.46±0.18 |
| ReinMax | **105.60** | **126.29±0.32** | **102.40±0.43** | **100.45±0.26** | **100.84±0.56** | **102.91±0.68** | 100.69±0.48 |

Table 11: Train −ELBO on MNIST. Hyper-parameters are chosen based on Test −ELBO.

|  | AVG | $8 \times 4$ | $4 \times 24$ | $8 \times 16$ | $16 \times 12$ | $64 \times 8$ | $10 \times 30$ |
|---|---|---|---|---|---|---|---|
| STGS | 105.31 | 126.85±0.85 | 101.81±0.14 | 99.32±0.33 | 100.22±0.47 | 104.02±0.41 | 99.63±0.63 |
| GR-MCK | 107.37 | 126.53±0.55 | 100.47±0.31 | 99.75±0.29 | 103.11±0.58 | 112.34±0.48 | 102.02±0.18 |
| GST-1.0 | 104.60 | 126.63±1.16 | 102.11±0.24 | 98.40±0.34 | 98.76±0.41 | 102.53±0.57 | 99.14±0.30 |
| ST | 117.76 | 136.75±0.22 | 112.09±0.50 | 113.06±0.26 | 113.31±0.43 | 113.90±0.28 | 117.46±0.09 |
| ReinMax | **103.40** | **124.92±0.38** | **99.77±0.45** | **98.06±0.31** | **98.51±0.54** | **100.71±0.70** | **98.40±0.48** |

**Differentiable Neural Architecture Search.** We adopt most of the hyper-parameter setting from Dong et al. (2020a). Since GDAS employs a temperature schedule (decaying linearly from 10 to 0.1), and temperature scaling works differently in ReinMax and STGS (as discussed in Section 5 and Section 6.6), we removed the temperature scaling (i.e., set the temperature to a constant 1.0) and increased the weight decay (i.e., from 0.001 to 0.09).

**ListOps.** We followed the same setting of Fan et al. (2022), i.e., used the same model configuration as in Choi et al. (2017) and set the maximum sequence length to 100.

### F.3 Hardware and Environment Setting

Most experiments (except efficiency comparisons) are conducted on Nvidia P40 GPUs. For efficiency comparisons, we measured the average time cost per batch and peak memory consumption on quadratic programming and MNIST-VAE on the same system with an idle A6000 GPU. Also, to better reflect the efficiency of gradient estimators, we skipped all parameter updates in efficiency comparisons.

### F.4 Additional Results on Polynomial Programming

We visualized the training curve for polynomial programming with various batch sizes and latent dimensions in Figure 8 (for $p = 1.5$), Figure 9 (for $p = 2$), and Figure 10 (for $p = 3$).

### F.5 Additional Results on MNIST-VAE

In our discussions in Section 6, we focused on the training ELBO only. Here, we provide a brief discussion on the test ELBO.

**Choosing Hyper-parameter Based on Training Performance.** Similar to Table 2, for each method, we select the hyper-parameter based on its training performance. The Test −ELBO in this setting is summarized in Table 9. Despite the model being trained without dropout or other overfitting reduction techniques, ReinMax maintained the best performance in this setting.

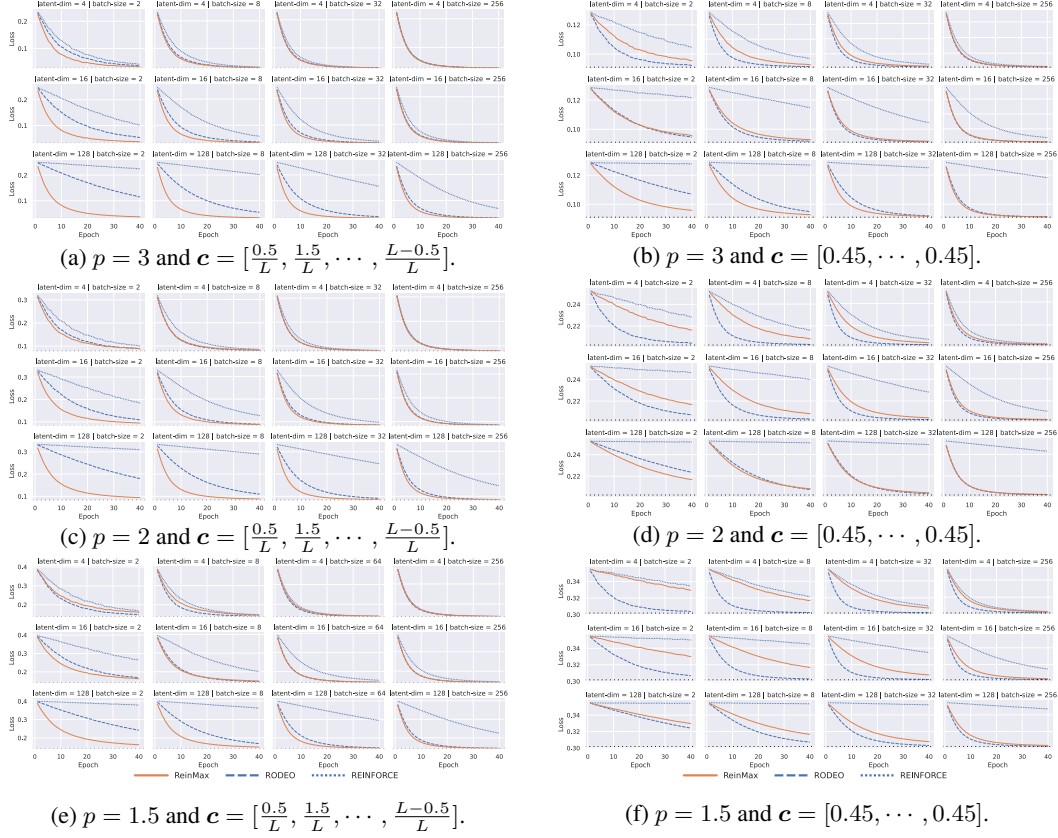

(a) $p = 3$ and $\boldsymbol{c} = [\frac{0.5}{L}, \frac{1.5}{L}, \cdots, \frac{L-0.5}{L}]$.

(b) $p = 3$ and $\boldsymbol{c} = [0.45, \cdots, 0.45]$.

(c) $p = 2$ and $\boldsymbol{c} = [\frac{0.5}{L}, \frac{1.5}{L}, \cdots, \frac{L-0.5}{L}]$.

(d) $p = 2$ and $\boldsymbol{c} = [0.45, \cdots, 0.45]$.

(e) $p = 1.5$ and $\boldsymbol{c} = [\frac{0.5}{L}, \frac{1.5}{L}, \cdots, \frac{L-0.5}{L}]$.

(f) $p = 1.5$ and $\boldsymbol{c} = [0.45, \cdots, 0.45]$.

Figure 7: Training curves of polynomial programming, i.e., $\min_{\boldsymbol{\theta}} E_{\boldsymbol{X}}[\frac{\|\boldsymbol{X}-\boldsymbol{c}\|_p^p}{L}]$, where $\boldsymbol{X} \in \{0,1\}^L$, $\boldsymbol{X}_i \overset{\text{iid}}{\sim}$ Multinomial(softmax($\boldsymbol{\theta}_i$)), $\boldsymbol{\theta} = [\theta_1, \cdots, \theta_L]^T$, $\boldsymbol{\theta}_i \in \mathbb{R}^2$, and $L$ is the number of latent dimensions.

**Choosing Hyper-parameter Based on Test Performance.** We also conduct experiments by selecting hyper-parameters directly based on their test performance. In this setting, the test $-$ELBO is summarized in Table 10, and the training $-$ELBO is summarized in Table 11. ReinMax achieves the best performance in all settings except the test performance of the setting with 10 categorical dimensions and 30 latent dimensions.

### F.6 More Comparisons with RODEO

To better understand the difference between RODEO and ReinMax, we conduct more experiments on polynomial programming, i.e., $\min_{\boldsymbol{\theta}} E_X[\frac{|\boldsymbol{X}-\boldsymbol{c}|_p^p}{L}]$. Specifically, we consider polynomial programming under two different settings that define $\boldsymbol{c}$ differently:

- In setting A, we have $\boldsymbol{c} = [0.45, \cdots, 0.45]$. This is the setting we used in the submission.

- In setting B, we have $\boldsymbol{c} = [\frac{0.5}{L}, \frac{1.5}{L}, \cdots, \frac{L-0.5}{L}]$.

As to the difference between the Setting A and the Setting B, we would like to note:

- In setting A, since $\forall i, \boldsymbol{c}_i = 0.45$ and $\boldsymbol{\theta}_i \sim$ Uniform$(-0.01, 0.01)$ at initialization, $E_{\boldsymbol{X}_i \sim \text{softmax}(\theta_i)}[\frac{|\boldsymbol{X}_i - \boldsymbol{c}_i|_p^p}{L}]$ would have similar values. Therefore, the optimal control variates for $\boldsymbol{\theta}_i$ are similar across different $i$.

- In setting B, we set $\boldsymbol{c}_i$ to different values for different $i$, and thus the optimal control variate for $\boldsymbol{\theta}_i$ are different across different $i$. Therefore, Setting A is a simpler setting for applying control variate to REINFORCE.

As in Figure 7, ReinMax achieves better performance in more challenging scenarios, i.e., smaller batch size, more latent variables, or more complicated problems (Setting B or VAEs). Meanwhile, REINFORCE and RODEO achieve better performance on simpler problem settings, i.e., larger batch size, fewer latent variables, or simpler problems (Setting A).

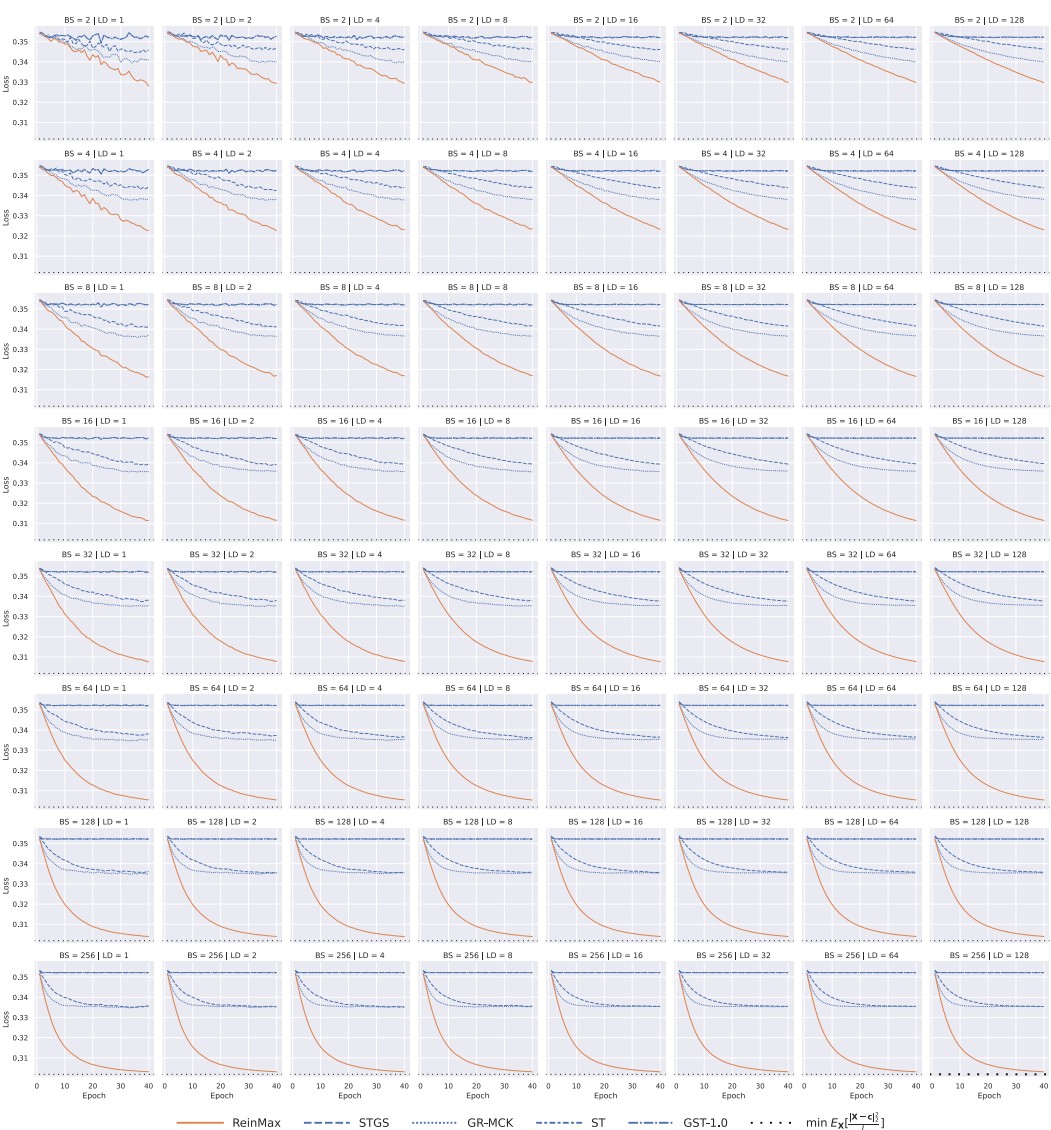

Figure 8: Polynomial programming training curve, with different batch sizes and random variable counts ($L$), i.e., $\min_{\boldsymbol{\theta}} E[\frac{\|\boldsymbol{X}-\boldsymbol{c}\|_{1.5}^{1.5}}{L}]$, where $\boldsymbol{\theta} \in \mathcal{R}^{L \times 2}$, $\boldsymbol{X} \in \{0,1\}^L$, and $\boldsymbol{X}_i \overset{iid}{\sim}$ Multinomial(softmax($\boldsymbol{\theta}_i$)). More details are elaborated in Section 6.

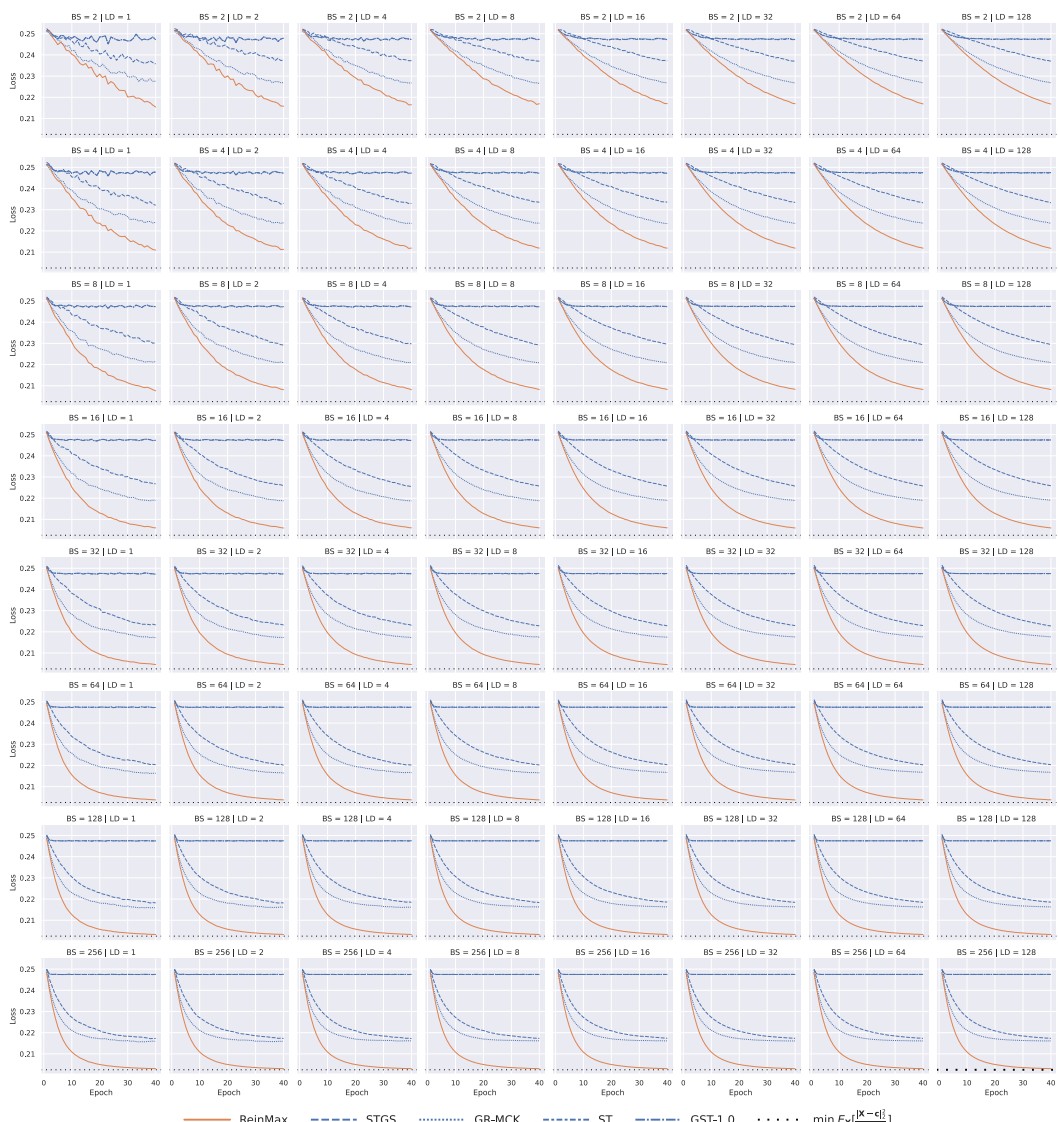

Figure 9: Quadratic programming training curve, with different batch sizes and random variable counts $(L)$, i.e., $\min_{\boldsymbol{\theta}} E[\frac{\|\boldsymbol{X}-\boldsymbol{c}\|_2^2}{L}]$, where $\boldsymbol{\theta} \in \mathcal{R}^{L\times 2}, \boldsymbol{X} \in \{0,1\}^L$, and $\boldsymbol{X}_i \stackrel{\text{iid}}{\sim}$ Multinomial(softmax($\boldsymbol{\theta}_i$)). More details are elaborated in Section 6.

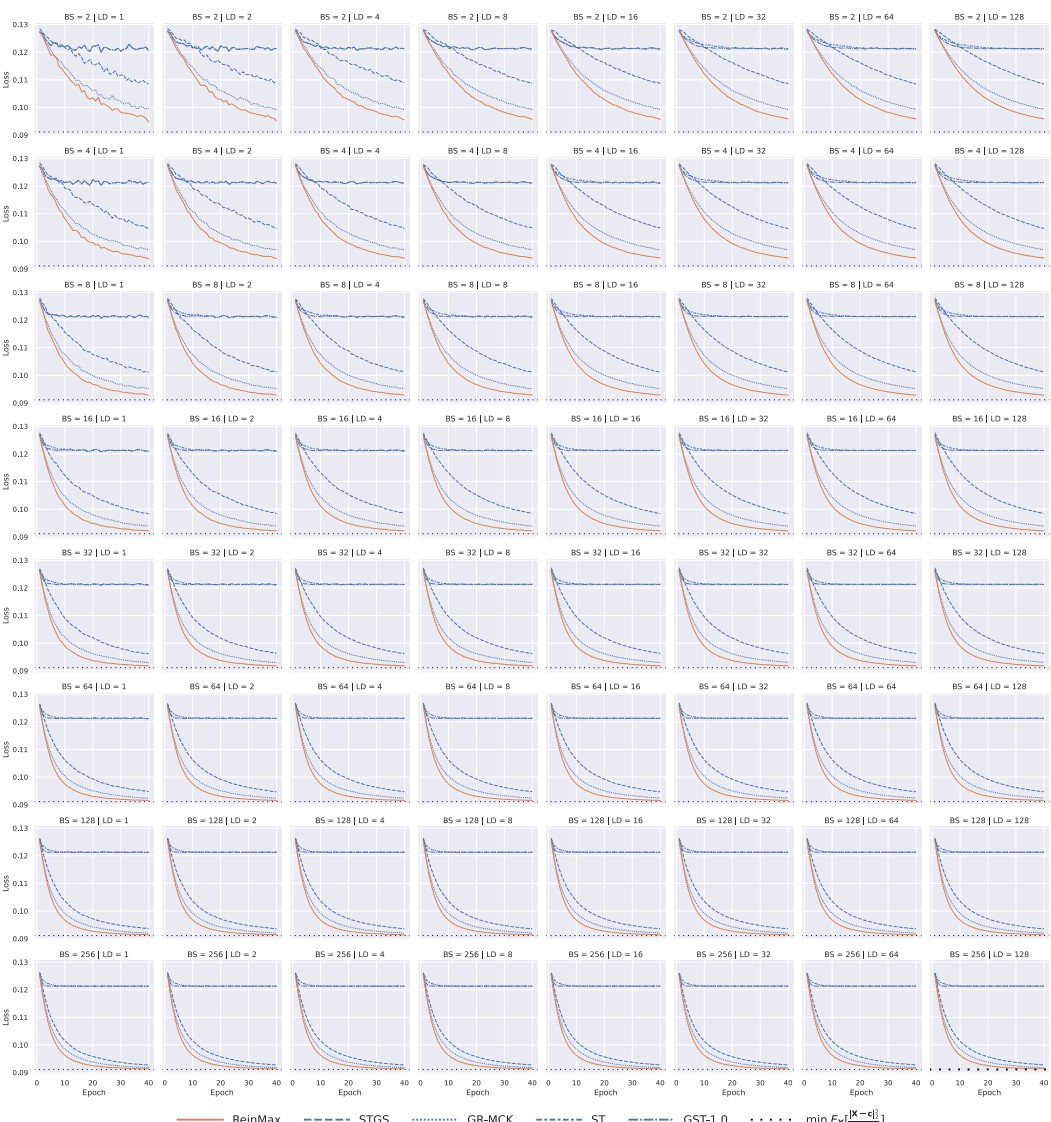

Figure 10: Polynomial programming training curve, with different batch sizes and random variable counts ($L$), i.e., $\min_{\boldsymbol{\theta}} E[\frac{\|\boldsymbol{X}-\boldsymbol{c}\|_3^3}{L}]$, where $\boldsymbol{\theta} \in \mathcal{R}^{L\times 2}, \boldsymbol{X} \in \{0,1\}^L$, and $\boldsymbol{X}_i \overset{\text{iid}}{\sim}$ Multinomial(softmax($\boldsymbol{\theta}_i$)). More details are elaborated in Section 6.

