# OpenReview forum: "Bridging Discrete and Backpropagation: Straight-Through and Beyond"
_NeurIPS.cc/2023/Conference — NeurIPS 2023 oral_

### Official Review · Reviewer_A6oc · 2023-07-07

**Soundness:** 3 good
**Presentation:** 3 good
**Contribution:** 3 good
**Rating:** 7
**Confidence:** 3

**Summary:**

The presented paper tackles the optimization of the parameter of the distribution of a discrete random variable through stochastic gradient descent by addressing the problem of gradient estimation given samples from the discrete distribution.
The authors consider three popular estimators as a starting point for their investigation, namely, the reinforce estimator, the straight-through estimator and the straight-through gumbel-softmax estimator.

The authors first establish that the straight-through estimator is a first-order approximation of the true gradient, since the difference $f(I_i) - f(I_j)$ in the true gradient is replaced by $\frac{\partial f(I_j)}{\partial I_j}(I_i - I_j)$.
The author thus replace the $\frac{\partial f(I_j)}{\partial I_j}(I_i - I_j)$, which gives a first-order approximation of $f(I_i) - f(I_j)$, by $\frac{1}{2} (\frac{\partial f(I_j)}{\partial I_j} + \frac{\partial f(I_i)}{\partial I_i})(I_i - I_j)$, which yield a second order approximation.

The authors then empirically evaluates both the baselines and the proposed estimator on a synthetic task (polynomial programming), unsupervised modeling (unsupervised sequence parsing), and structured output prediction (generative modeling).



**Strengths:**

- The theoretical analysis is well written and intuitively explained.
The interpretation of the straight-through estimator as a first-order approximation of the true gradient allows the author to use numerical analysis results to improve the theoretical behavior of their proposed solution.
- The experiment section is decent, with qualitatively different setups addressing different application domains.
- The choice of the baseline for the baseline subtraction method as well as the tuning of the temperature parameter is appropriately discussed.

**Weaknesses:**

- Equation (6) does not seem to be valid. I would expect $\sum_i (f(I_i) - E[f(D)]) \frac{d \pi_i}{d\theta} = \sum_i \sum_j \pi_j(f(I_i) - f(I_j)) \frac{d \pi_i}{d\theta}$, thus I don't understand why the term $\sum_i E[f(D)] \frac{d \pi_i}{d \theta}$ vanishes.
- The derivation of remark 4.1 as provided in appendix C does not seem to be valid. I don't understand how
$\sum_i \sum_j \pi_j \frac{\partial f(I_j)}{\partial I_j}(I_i - I_j)) \frac{d \pi_i}{d\theta} = \sum_i \frac{\phi_D}{\pi_D} \pi_D \frac{\partial f(I_j)}{\partial I_j} \sum_i I_i \frac{d \pi_i}{d\theta}$. Furthermore, there should be a typo at line 155 since it should be $\pi_D$ who is the output of the softmax and could take very small values. This would be consistent with the fact that it is the denominator of the fraction $\frac{\phi_D}{\pi_D}$.
- I have no idea how equation (8) has been derived.

**Questions:**

It is said that the derivation of equation (8) leverages the derivative of the softmax function, thus there may be a handy formula about these derivative that I am not aware of and which justify all the equations that do not seem valid to me. Could the authors elaborate on my concern about the validity of their equations ?

I highly doubt that the equations are wrong, given that they coincide with previous work on the topic and the consistent improvements provided in the experiment section. I would be willing to increase my score if the author correctly addresses my concerns.

**Limitations:**

There is a decent discussion about the limitation of the proposed method througout the paper.
Albeit, given that high variance is the main limiting factor of the REINFORCE estimator, and given that the other estimators are biased, a focus on bias variance tradeoff would have been welcome.

---

> ### Author Rebuttal · Authors · 2023-08-10
>
> Thank you for your constructive feedback. We value your comments and will address your concerns regarding the correctness of the derivation, with further elaborations to be included in the final paper.
>
>
> **Reply to weakness argument 1:** Since $\sum_i \pi_i = 1$, we have $\sum_i E[f(D)] \frac{d \pi_i}{d \theta} = E[f(D)] \cdot \frac{d\sum_i \pi_i }{d\theta} = E[f(D)] \cdot \frac{d 1}{d\theta} = 0$
> In the revision, we will add a simple explanation of this in Eq. (6).
>
> **Reply to weakness argument 2:** Thanks for pointing out the typo, and we will fix the typo and add more elaborations in the revision. Please find the detailed derivation of the last-second equation in Appendix C as below.
>
>
> $\sum_i \sum_j \phi_j \frac{\partial f(I_j)}{\partial I_j} (I_i - I_j) \frac{d \pi_i}{d \theta}$
>
> $= \sum_j \Large(\normalsize \phi_j\frac{\partial f(I_j)}{\partial I_j} \sum_i (I_i - I_j) \frac{d \pi_i}{d \theta} \Large)\normalsize$
>
> $= \sum_j \Large(\normalsize \phi_j\frac{\partial f(I_j)}{\partial I_j} (\sum_i I_i \frac{d \pi_i}{d \theta} - \sum_i I_j \frac{d \pi_i}{d \theta} )\Large)\normalsize$
>
> $= \sum_j \Large(\normalsize \phi_j\frac{\partial f(I_j)}{\partial I_j} (\sum_i I_i \frac{d \pi_i}{d \theta} - I_j \frac{d \sum_i \pi_i}{d \theta} )\Large)\normalsize$
>
> $= \sum_j \Large(\normalsize \phi_j\frac{\partial f(I_j)}{\partial I_j} (\sum_i I_i \frac{d \pi_i}{d \theta} - I_j \frac{d 1}{d \theta}) \Large)\normalsize$
>
> $= \sum_j \phi_j\frac{\partial f(I_j)}{\partial I_j} \sum_i I_i \frac{d \pi_i}{d \theta}$
>
> $= \sum_j \frac{\phi_j}{\pi_j}\cdot \pi_j \cdot \frac{\partial f(I_j)}{\partial I_j} \sum_i I_i \frac{d\pi_i}{d\theta}$.
>
>
> **Reply to weakness argument 3 and question 1:**
> In the revision, we will mention the derivative of the softmax around L171, i.e., for $\pi = \mbox{softmax}(\theta)$, we have $\partial \pi_i / \partial \theta_k = \pi_k (\delta_{ik} - \pi_i)$. Please find the detailed derivation of equation (8) as below.
>
> $\frac{\partial \mathcal{L}}{\partial \theta_k}= \frac{\partial \sum_i \pi_i f(I_i) }{\partial \theta_k}$
>
> $=\sum_i f(I_i) \frac{d \pi_i}{d \theta_k}$
>
> $= \sum_i f(I_i) \pi_k (\delta_{ik} - \pi_i)$
>
> $= \sum_i f(I_i) \pi_k \delta_{ik} - \sum_i f(I_i) \pi_k \pi_i$
>
> $= f(I_k) \pi_k - \sum_i f(I_i) \pi_k \pi_i$
>
> $= f(I_k) \pi_k \sum_i \pi_i - \sum_i f(I_i) \pi_k \pi_i$
>
> $= \pi_k \sum_i \pi_i (f(I_k) - f(I_i))$.
>
> We hope our responses have adequately addressed your concerns and further highlighted the innovations and potential impact of our study. If you have any further questions or need additional information, please do not hesitate to ask.

---

> > ### Comment · Reviewer_A6oc · 2023-08-16
> >
> > I thank the authors for taking the time to carefully right each step of their derivation to me. As other reviewers suggested, some details needed for the full derivation should be mentioned in the main text, or I would alternatively suggest to include the derivations in the appendix if the main body run out of space.
> >
> > Since the proposed contribution consistently outperforms alternative methods in the experiments provided in the paper as well as the additional experiments provided in the global rebuttal, I am increasing my score accordingly.

---

> > > ### Author Response · Authors · 2023-08-17
> > > **Author Response**
> > >
> > > Thank you for your feedback. We appreciate your acknowledgment of our work's performance in the experiments, and we will include the discussions and clarifications here.

---

### Official Review · Reviewer_fihA · 2023-07-09

**Soundness:** 3 good
**Presentation:** 3 good
**Contribution:** 3 good
**Rating:** 7
**Confidence:** 4

**Summary:**

This paper works on gradient approximation for discrete latent variables. The problem is challenging because the discreteness hinders direct backpropagation through the neural networks; hence, an approximation is needed. The authors address the issues from the perspective of a second-order approximation of the gradient. Specifically, the authors proposed ReinMax and showed that ReinMax achieved second-order accuracy with negligible computation overheads. The experiments on Polynomial Programming, MNIST-VAE, and ListOps demonstrated the superiority of ReinMax over the prior methods such as Straight-Through Gumbel-Softmax, Gumbel-Rao Monte Carlo, and Gapped Straight-Through.

**Strengths:**

The problem has been challenging for a long time. Most prior work addresses the issue from the perspective of Straight-Through Gumbel Softmax (STGS). To the reader's best knowledge, the perspective of second-order approximation is novel in this field and the performance improvement is believable. Below highlights the strengths of the paper.

a) The use of second-order approximation is novel and the theoretical derivation is believable.

b) The performance gain is clear with similar computational overhead as in the prior work.

c) The experimental analysis covers some relevant topics such as temperature scaling and the choice of baseline. The conclusion is clear from the analysis.

**Weaknesses:**

a) The design choice of ReinMax is unclear.

Although ReinMax is well-motivated by the second-order approximation, the reader might be confused by the form of ReinMax at Eq. (7). For example, why mixing $\pi$ and $D$ by introducing $\pi_D=(\pi+D)/2$? Why substracts by $\nabla_{ST}$? Where are the coefficients (2 and 1/2) from? It would be nice if the authors can elaborate more on these.

b) The implication of the experiments remains unclear.

For a fair comparison with the prior work, the authors follow the literature and run ReinMax on Polynomial Programming, MNIST-VAE, and ListOps. However, these are toy problems with limited implications for the applications. For example, does the good performance of ReinMax on Polynomial Program/MNIST-VAE/ListOps imply any possible applications? Conversely, if we have a reinforcement learning problem with discrete actions or a VQ-VAE model, can ReinMax be helpful? It would be nice if the authors can evaluate more on the application side so that the other practitioners can follow.

**Questions:**

Why the proposed method is called ReinMax?

For the other questions, see the weakness section.

**Limitations:**

No potential negative social impact.

---

> ### Author Rebuttal · Authors · 2023-08-10
>
> Thank you for your constructive feedback. We value your comments and will address the concerns regarding experimental design and presentation in this rebuttal, with further elaborations to be included in the final paper.
>
> **Reply to weakness 1:** We understand the complexity of ReinMax's exact form may not be immediately intuitive, as it is a result of specific derivations. To alleviate confusion, we will dedicate additional space in the final version to elaborate on the exact form of ReinMax.
>
> **Reply to weakness 2:** In our submission, we adhered to the experiment design of the existing study, focusing on small-scale problems for controllability and resource efficiency, as our experiments were mainly conducted on P100/P40 GPUs.
>
> To further demonstrate the generalizability of ReinMax, we conducted additional experiments (detailed in the general rebuttal), including (1) comparisons with the REINFORCE variant that employs the state-of-the-art variance reduction technology, specifically RODEO (SHI 2022), and (2) applications to a real-world scenario, i.e., differentiable neural architecture search on CIFAR10, CIFAR100, and ImageNet-16-120.
>
> ReinMax maintained outstanding performance throughout these expanded tests, showing consistent improvements over the baseline. These additional experiments should provide a more comprehensive understanding of ReinMax's potential applications.
>
> **Reply to question 1:** We used the paper title and the abstract as the prompt and queried ChatGPT for several names, among which we selected ReinMax, since it concisely implies the method's connection to softmax and REINFORCE, and we believe it is a fitting and appealing term.
>
> We hope our responses have adequately addressed your concerns and further highlighted the innovations and potential impact of our study. If you have any further questions or need additional information, please do not hesitate to ask.

---

> > ### Comment · Area_Chair_6AcV · 2023-08-18
> > **Thanks**
> >
> > Thank you for this feedback authors. This will be taken into account.

---

### Official Review · Reviewer_txhR · 2023-07-09

**Soundness:** 4 excellent
**Presentation:** 4 excellent
**Contribution:** 3 good
**Rating:** 7
**Confidence:** 4

**Summary:**

The paper proves that the straight-through (ST) gradient estimator for the categorical distribution can be viewed as a first-order approaximation of the true gradients. Based on this point of view, a new gradient estimator, ReinMax is proposed based on Heun's method which is a second-order method. Experiments are conducted which show that ReinMax improves upon the state of the art methods.

**Strengths:**

- **The paper has good writing quality.** The background is introduced in a well-organized manner and contributions are stated clearly. Mathematical formulae are also well laided out and are easy to understand. Relationship with existing work (ST, REINFORCE, baseline subtraction) are also expanded upon in good detail and nuance.

- **Solid contribution with a comprehensive set of experiments.** The proposed new method has solid theoretical underpinnings (Heun's method being more accurate than Euler's method), is still very efficient (not requiring the Hessian), and is proven to be effective under multiple different settings (polynomial programming, MNIST, etc.). The new method is compared against many commonly used methods (STGS, DisARM, ...) and there is also discussion of using a different baseline for subtraction. Extensive hyperparameter tuning is performed and advantage of the proposed method is demonstrated well.

**Weaknesses:**

My main concern with this paper is that it **oversells its theoretical contribution**. The view of the straight-through estimator as a first-order Taylor expansion of the unbiased gradients is not news to people who study gradient estimators. The authors are fair in pointing out that previous work (Tokui & Sato 2017) only dealt with the Bernoulli case, but one might argue that The categorical distribution is just a natural extension of the Bernoulli case. Regardless, I think it is **potentially misleading to say that "ST as a first-order approximation to the gradient" is a novel perspective**. Despite this, I still consider the proposed approach to be novel enough that the contribution of the paper as a whole is still significant.

Another concern is the lack of comparison with RODEO (Shi 2022) which as far as I know is the state of the art for REINFORCE methods with variance reduction. The paper only reported numbers on RLOO and DisARM in table 3 but these methods are known to be weaker than RODEO, which is reported to outperform DisARM significantly in terms of gradient variance.

**Questions:**

- What is the effect of temperature scaling on the proposed method? How does it influence the relative performance of ST-type methods when compared with unbiased REINFORCE-based methods?
- Is there a complexity-efficiency trade-off here? For example, would using 4th-order Runge-Kutta instead of Heun's method yield much better gradient estimations and improve the overall result?

**Limitations:**

The authors do not explicitly discuss the limitations of their work.

---

> ### Author Rebuttal · Authors · 2023-08-10
>
> Thank you for your constructive feedback. We value your comments and will address the concerns regarding the novelty of our study and experiment design, with further elaborations to be included in the final paper.
>
> **Reply to weakness argument 1:**  While it may be intuitive to some that the straight-through estimator functions as a gradient approximation, no prior work has formally established this for the general multinomial case.
>
> Also, the derivation of Theorem 3.1, while not overly complicated, is more than a mere expansion of existing results:
>
> - In Tokui & Sato (2017), the authors positioned $\hat{\nabla}_{ST}$ as a first-order approximation, but their analysis is exclusively rooted in the properties of Bernoulli variables. As an example, let us consider a Bernoulli random variable $D \\in \\{ I_1, I_2 \\}$. Their approach depends on the property that $\\nabla = (f(I_2) - f(I_1)) \frac{d\\pi_1}{d \\theta} = (f(I_1) - f(I_2)) \\frac{d\\pi_2}{d \\theta}$, and thus is not applicable to multinomial variables.
>
> - On the other hand, the analyses in Gregor et al. (2014) and Pervez et al. (2020) are applicable to multinomial variables but resort to adding the term $\\frac{1}{n \\cdot \\pi_D}$ in $\\hat{\\nabla}_{ST}$, an alteration that we believe could induce unwanted instability. This concern is discussed in Section 4.1 and Section 6.4.
>
> In the revision, we will modify our claim as “our study is the first to formally establish $\\hat{\\nabla}_{ST}$ works as a first-order approximation in the general multinomial case".
>
> **Reply to weakness argument 2:**
> In the general rebuttal, we provide additional experimental results on  (1) comparisons with the REINFORCE variant that employs the state-of-the-art variance reduction technology, specifically RODEO (SHI 2022), and (2) applications to a real-world scenario, i.e., differentiable neural architecture search on CIFAR10, CIFAR100, and ImageNet-16-120.
>
> ReinMax maintained outstanding performance throughout these expanded tests, showing consistent improvements over the baseline.
>
>
> **Reply to question 1:**
> In our experiments, we observed that temperature scaling enhances the stability of both ST and ReinMax algorithms. We conjecture that this scaling acts as a variance reduction method.
>
> Across all six VAE settings discussed in the general rebuttal, ReinMax achieved the best performance when the temperature was set to 1.1. Adjusting the temperature to 1.0 or 1.2 would lead to a slight performance drop (with an ELBO difference of approximately 1), while ReinMax still outperformed RODEO with these sub-optimal temperatures.
>
> **Reply to question 2.1**
> As discussed in Section 6.4, the computation overhead of ReinMax is negligible.
>
> **Reply to question 2.2**
> Although it's possible to apply higher-order ODE solvers, they require more gradient evaluations, leading to undesirable computational overhead. To illustrate this point:
> - The approximation used by ReinMax (as described in Definition 3.2) requires N gradient evaluations, i.e., $\\{\\frac{\\partial f(I_i)}{\\partial I_i}\\}$.
> - In contrast, the approximation derived by RK4 needs $N^2+N$ gradient evaluations, i.e., $\\{\\frac{\\partial f(I_i)}{\\partial I_i}\\}$ and $\\{\\frac{\\partial f(I_{ij})}{\\partial I_{ij}}\\}$, where $I_{ij} = \\frac{I_i + I_j}{2}$.
>
> Therefore, while higher-order solvers are applicable, they may not be suitable in our case.
>
> We hope our responses have adequately addressed your concerns and further highlighted the innovations and potential impact of our study. If you have any further questions or need additional information, please do not hesitate to ask.

---

> > ### Comment · Reviewer_txhR · 2023-08-16
> >
> > I thank the authors for their response and for providing further insights on the theoretical aspects of their work. The authors added comparison with the state of the art method for REINFORCE variance reduction and provided insight on the strengths of each method which I found to be helpful for my understanding of their work. I am also happy to know that the authors have acknowledge my point on the novelty of the theoretical perspective and are willing to adjust the writing. With the above concerns addressed, I am bumping up my score to a 7.

---

> > > ### Author Response · Authors · 2023-08-17
> > > **Author Response**
> > >
> > > Thank you for your constructive feedback. We're glad our clarifications were helpful. we will include the discussions and clarifications in the revision.

---

### Official Review · Reviewer_ZCX1 · 2023-07-12

**Soundness:** 3 good
**Presentation:** 3 good
**Contribution:** 3 good
**Rating:** 7
**Confidence:** 2

**Summary:**

This work develops a new method, ReinMax, to compute gradients of parameters
used to generate discrete random variables; specifically, the parameters of a
multinomial's softmax parameterization. It provides a new perspective onto the
existing straight-through (ST) gradient estimator as a first-order
approximation. The new method is a second-order approximation and, in contrast
to ST, relies on gradient differences rather than a single gradient. The ReinMax
gradient estimator is compared to other approaches on polynomial programming toy
problems, unsupervised parsing on ListOps, and ELBO training of a VAE on MNIST.


**Strengths:**

- (S1) The paper provides a new perspective onto the Straight-Through (ST) technique,
  which simply replaces the backpropagation through a non-differentiable
  operation by applying the identity. This adds a new theoretical justification
  why such heuristics work. Framing ST as first-order approximation also naturally
  suggests an extension to second-order, as presented by the paper.

- (S2) The new method does not seem to add significant overhead to existing
  approaches. The analysis includes a sensitivity analysis over different
  problem hyper-parameters which suggests that the estimation works
  consistently.


**Weaknesses:**

- (W1) Limited to softmax parameterization of multinomial: The main text focuses
  on computing gradients for the parameters of a multinomial parameterized by a
  softmax. The authors should comment on the (im)possibility to treat other
  parameterizations or distributions.

- (W2) Experiments: In the presented experiments, ReinMax seems to consistently
  work well and perform comparably or more favourably than the competitors.
  However, I do not have the expertise to judge whether the experiments
  represent state-of-the-art tasks. I am willing to adapt my score and
  confidence during the discussion phase with the authors and other reviewers.

- (W3) Clarity: Some steps could be easier to follow by providing the required
  mathematical properties. The figures and tables should be moved closer to
  where they are referenced in the text. Here is a list of actionable
  suggestions which I think would improve clarity:
  - In Eq. (4), explicitly add the term $\partial D / \partial \pi$ that is set
    to the identity in ST
  - In Eq. (6), mention that $\sum_i \partial \pi_i / \partial \theta = \partial
    \sum_i \pi_i / \partial \theta = \partial 1 / \partial \theta = 0$.
  - I think it would be helpful to move parts of appendix E to the main text to
    have a more formal description of first- and second-order approximation in
    the specific context.
  - Fix '2rd-order' into '2nd-order' in various places in the main text
    and the appendix.
  - Mention the derivative of a softmax around L171, that is $\partial \pi_i /
    \partial \theta_k = \pi_k (\delta_{ik} - \pi_i)$
   - Clarify the notation $\delta_{\mathbf{D} k}$ in the appendix
  - Minor suggestions: Use consistent symbol $\mathcal{R}$ in L51, 'softmax' in
    caption of Figure 1, remove 'on' in L84, 'computational efficiency' rather
    than 'computation efficiency', 'approximation' in L185, use bold symbol for
    $\theta_i$ in L210 and add $\in \mathcal{R}^2$, 'phenomena' rather than
    'phenomenon' in L218), add 'that' between 'one' and 'uses' in L264



**Questions:**

- Why is the method called ReinMax?

- ReinMax is a second-order extension of ST. Can this be extended to even higher
  orders? Would this be beneficial or are there diminishing returns in including
  higher orders?

- What is $\phi_i$ in L150 and what are its constraints?


**Limitations:**

See  (W1)

---

> ### Author Rebuttal · Authors · 2023-08-10
>
> Thank you for your constructive feedback. We value your comments and will address the concerns regarding experimental design, presentation, and limitations of ReinMax in this rebuttal, with further elaborations to be included in the final paper.
>
> **Reply to limitation 1:** While our main text focuses on multinomial distributions, by reparameterizing other categorical distributions into a multinomial distribution, ReinMax can be generally applied to all categorical random variables.
>
> It is worth mentioning that ReinMax is not applicable to continuous random variables. While extending ReinMax to these cases is possible, it may not be necessary, given that many commonly used distributions can be reparameterized. For example, the normal distribution $z \sim \mathcal{N}(\mu, \sigma)$ can be re-written as $z = \mu + \sigma \cdot \mathcal{N} (0, 1)$, making it trivial to compute $\partial z/\partial \mu$ and $\partial z/\partial \sigma$.
>
> Furthermore, ReinMax differs from REINFORCE in that ReinMax requires both the $\frac{\partial f(D)}{\partial D}$ and $\frac{\partial p(D)}{\partial \theta}$, while REINFORCE only requires $\frac{\partial p(D)}{\partial \theta}$. Thus, REINFORCE can be applied in cases where $f(D)$ is not differentiable.
>
> **Reply to weakness 2:** In our submission, we adhered to the experiment design of the existing study, focusing on small-scale problems for controllability and resource efficiency, as our experiments were mainly conducted on P100/P40 GPUs.
>
> To further demonstrate the generalizability of ReinMax, we conducted additional experiments (detailed in the general rebuttal), including (1) comparisons with the REINFORCE variant that employs the state-of-the-art variance reduction technology, specifically RODEO (SHI 2022), and (2) applications to a real-world scenario, i.e., differentiable neural architecture search on CIFAR10, CIFAR100, and ImageNet-16-120.
>
> Our findings showed that ReinMax performs strongly and consistently in these settings. We will add more discussions and elaborations about these experiments in the final version of the paper.
>
> **Reply to weakness 3:** We appreciate your suggestions for improving clarity, and we will incorporate them into our revision.
>
> **Reply to question 1:** We used the paper title and the abstract as the prompt to query ChatGPT for several names, among which we selected ReinMax since it concisely implies the method's connection to softmax and REINFORCE, and we believe it is a fitting and appealing term.
>
> **Reply to question 2** Although it's possible to apply higher-order ODE solvers, they require more gradient evaluations, leading to undesirable computational overhead. To illustrate this point:
> - The approximation used by ReinMax (as described in Definition 3.2) requires N gradient evaluations, i.e., $\\{\\frac{\\partial f(I_i)}{\\partial I_i}\\}$.
> - In contrast, the approximation derived by RK4 needs $N^2+N$ gradient evaluations, i.e., $\\{\\frac{\\partial f(I_i)}{\\partial I_i}\\}$ and $\\{\\frac{\\partial f(I_{ij})}{\\partial I_{ij}}\\}$, where $I_{ij} = \\frac{I_i + I_j}{2}$.
>
> Therefore, while higher-order solvers are applicable, they may not be suitable in our case.
>
> **Reply to question 3** $\phi$ is a distribution over $\{I_1, \cdots, I_n\}$, i.e., $\sum_i \phi_i = 1$ and $\phi_i = P(I_i)$.
>
> We hope our responses have adequately addressed your concerns and further highlighted the innovations and potential impact of our study. If you have any further questions or need additional information, please do not hesitate to ask.

---

> > ### Comment · Reviewer_ZCX1 · 2023-08-14
> > **Rebuttal follow-up**
> >
> > Dear authors,
> >
> > thanks for the rebuttal. I am satisfied with your discussion of ReinMax's applicability/limitations (W1) and the possible extension to higher-order estimators. **Please make sure to include them in the draft**.
> >
> > You should also include a footnote that explains the algorithm's name, as this was brought up by multiple reviewers.
> >
> > Based on the overall positive feedback on your experiments (W2) from the other reviewers, as well as the additional results you provided in the rebuttal, I have decided to raise my score.

---

> > > ### Author Response · Authors · 2023-08-14
> > > **Author response**
> > >
> > > Thanks again for your timely response and detailed suggestions! We will include these discussions and additional results in the revision.

---

### Official Review · Reviewer_WPn8 · 2023-07-20

**Soundness:** 4 excellent
**Presentation:** 3 good
**Contribution:** 4 excellent
**Rating:** 8
**Confidence:** 4

**Summary:**

This paper is about designing a new approach to approximating the gradient of parameters in generating discrete random variables.
The work first starts with discovering the connection of Straight-Through (ST) estimator and the first-order approximation of the true gradient.
From the insight, the authors aim to improve the ST estimator, by applying second-order approximation of the true gradient. To apply second-order approximation without actually calculating second-order derivatives, the authors use Heun's Method, which approximates second-order derivatives using two first-order derivatives; thus no expensive calculation is needed.
The proposed estimator is coined ReinMax.
Through mathematical analyses, the effectiveness of using the expected value of function outputs is proven.
From evaluations, the authors empirically prove ReinMax method outperforms other estimators, along with presenting other properties and insights e.g. sensitivity to the number of dimensions, batch size, convergence speed, memory usage, and running time.

**Strengths:**

- The paper is well-organized and easy to follow
- Mathematical background is given thoroughly
- The method can be applied with minimal change of code

**Weaknesses:**

- Equation 6 seems to be one of core findings from the paper; the derivation process may be written more comprehensibly (like in Appendix A)
- More evaluations on benchmark datasets close to real-world distribution would be beneficial. For instance, how much improvement will be made when we apply the ReinMax estimator into other models for NLI or sentiment analysis instead of ListOps?

**Questions:**

- In experiments using the most basic REINFORCE algorithm, not RLOO or DisARM-Tree, is there any baseline subtraction used? If not can the $E[f(I_i)]$ baseline be used also in REINFORCE?

**Limitations:**

As mentioned in the checklist, I suppose no potential negative societal impact will arise by this work. Experiments are done quite fairly, including standard deviations of scores and implementation code.

---

> ### Author Rebuttal · Authors · 2023-08-10
>
> Thank you for your constructive feedback. We value your comments and will address the concerns regarding experimental design in this rebuttal, with further elaborations to be included in the final paper.
>
> **Reply to weakness 1:** Thanks for the suggestions! We will add more elaborations in the revision.
>
> **Reply to weakness 2:** In our submission, we adhered to the experiment design of the existing study, focusing on small-scale problems for controllability and resource efficiency, as our experiments were mainly conducted on P100/P40 GPUs.
>
> To further demonstrate the generalizability of ReinMax, we conducted additional experiments (detailed in the general rebuttal), including (1) comparisons with the REINFORCE variant that employs the state-of-the-art variance reduction technology, specifically RODEO (SHI 2022), and (2) applications to a real-world scenario, i.e., differentiable neural architecture search on CIFAR10, CIFAR100, and ImageNet-16-120.
>
> ReinMax maintained outstanding performance throughout these expanded tests, showing consistent improvements over the baseline.
>
> **Reply to question 1:** In the general rebuttal, we detailed comparisons between ReinMax and RODEO, a REINFORCE variant employing a state-of-the-art variance reduction method. RODEO outperforms ReinMax on simple scenarios (e.g., large batch size, small number of latent variables, Setting A). Meanwhile, ReinMax achieves better performance on complex scenarios (e.g., small batch size, large number of latent variables, VAE, and Setting B). We will add more elaborations and make corresponding revisions in the final version.
>
> We hope our responses have adequately addressed your concerns and further highlighted the innovations and potential impact of our study. If you have any further questions or need additional information, please do not hesitate to ask.

---

> > ### Comment · Area_Chair_6AcV · 2023-08-18
> > **Thanks**
> >
> > Thank you for this feedback authors. This will be taken into account.

---

### Official Review · Reviewer_GD4h · 2023-07-20

**Soundness:** 4 excellent
**Presentation:** 4 excellent
**Contribution:** 3 good
**Rating:** 7
**Confidence:** 3

**Summary:**

The paper shows that the straight-through estimator is a first-order approximation of the gradient. The authors then propose a method, called ReinMax, which provides a second-order approximation with negligible computational overhead. Experiments are performed in several settings involving discrete variables (polynomial programming, structured output prediction, and discrete latent variable generative models), showing that ReinMax is more accurate and stable.

**Strengths:**

**Originality**

The originality of the paper is fairly strong. As I see it, the main contributions of the paper are in 1) identifying the straight-through estimator as an instance of the forward Euler method, which is a first-order approximation of the gradient, and 2) using Heun’s method to derive a second-order gradient approximation. To someone mostly outside of the field of gradient approximation, these are non-trivial insights that, to the best of my knowledge, are unique to this paper.

**Quality**

The paper is high quality. I particularly appreciated the effort that the authors put into the empirical evaluation, performing hyperparameter sweeps to demonstrate stability, as well as empirically verifying that the ReinMax gradient estimator provides improved estimates of the true gradient. Overall, the result is a paper that provides compelling evidence of 1) improved understanding of straight-through and 2) an improved gradient estimator.

**Clarity**

Overall, the paper is quite clear. The authors do an excellent job of presenting mathematical notation to help the reader see the similarities across gradient derivations and estimators. Theoretical results are presented in a clear, logical order. The empirical results are also generally presented well, with clear labeling of tables and plots.

**Significance**

While many papers have mentioned that the straight-through estimator is an approximation of the gradient, it appears that none of these papers have formally shown it. If, indeed, this paper is the first to do so, then it is a significant contribution to our theoretical understanding. The proposed improved estimator, ReinMax, also appears to offer some performance improvement over previously proposed (first-order) estimators. Considering that ReinMax has similar computational overhead as straight-through, then this could serve as a drop-in replacement for the straight-through estimator throughout various applications. However, it is unclear to me whether this serves as a drop-in replacement for all existing instances of the straight-through estimator, or merely those that operate on Multinomial distributions.

**Weaknesses:**

I see two relatively minor weaknesses in the paper as-is: larger-scale empirical evaluation and slight improvements to the presentation. These are discussed below.

The current empirical evaluation involves three settings: quadratic programming, structured output prediction, and latent variable generative modeling with discrete latent variables. While the authors demonstrate ReinMax in all three settings, much of the empirical evaluation revolves around the final setting (categorical VAE on MNIST). These results are a useful indicator of the benefits of ReinMax, however, the empirical setting itself is rather toy-ish compared with modern settings. Given that ReinMax is a drop-in replacement for the straight-through estimator, I would imagine that it should be trivial to replace the ST estimator in an existing scaled-up setting, e.g., vector-quantized VAEs or Hafner et al.’s discrete world models. The authors could alternatively / additionally explore categorical VAEs on more complex data. This would allow the authors to more definitively claim that their proposed gradient estimator leads to tangible empirical improvements.

Additionally, several minor aspects of the presentation could be improved.
* The plots in Figure 1 are presented with fairly minimal context in the surrounding text. The caption states “Details are elaborated in Section 6.” Then it may make sense to place this figure closer to Section 6.
* As far as I can tell, the name “ReinMax” is never actually explained.
* The labels for the baseline methods, e.g., in Figure 1, 4, etc. are difficult to read; they are various dashed lines. Colors and/or larger lines would make this clearer.
* The shaded regions in Figure 5 (right) are rather jagged — perhaps there’s an issue with the evaluation interval or the plotting setup.
* In various tables, e.g., Tables 1, 2, …, the results for ReinMax are bolded, despite falling within the error bounds of the baseline methods. I find this to be somewhat misleading.
* The citation in the second sentence of the introduction is incorrect.

**Questions:**

Could you please elaborate on the applicability of the ReinMax estimator? Does this estimator (or the insights developed in the theoretical sections of the paper) alleviate the “laborious and time-consuming” work of developing “different ST variants for different applications in a trial-and-error manner”?

**Limitations:**

As mentioned above, I would appreciate a clearer discussion of which existing forms of straight-through estimator the ReinMax estimator can replace.

---

> ### Author Rebuttal · Authors · 2023-08-10
>
> Thank you for your constructive feedback. We value your comments and will address the concerns regarding experimental design, applicability, and limitations of ReinMax in this rebuttal, with further elaborations to be included in the final paper.
>
> **Reply to Weakness 1 and Question 1:** In our submission, we adhered to the experiment design of the existing study, focusing on small-scale problems for controllability and resource efficiency, as our experiments were mainly conducted on P100/P40 GPUs.
>
> To further demonstrate the generalizability of ReinMax, we conducted additional experiments (detailed in the general rebuttal), including (1) comparisons with the REINFORCE variant that employs the state-of-the-art variance reduction technology, specifically RODEO (SHI 2022), and (2) applications to a real-world scenario, i.e., differentiable neural architecture search on CIFAR10, CIFAR100, and ImageNet-16-120.
>
> In the architecture search application, ReinMax consistently improved performance over the baseline in a plug-and-play manner by:
> - Replacing STGS with ReinMax as the gradient estimator,
> - Conducting a minor change to the temperature hyper-parameters (changing the minimal value of the temperature from 0.1 to 1.1), as guided by our findings (refer to Sections 5 and 6.2).
>
> **Reply to limitation 1:** While our main text focuses on multinomial distributions, by reparameterizing other categorical distributions into a multinomial distribution, ReinMax can be generally applied to all categorical random variables.
>
> It is worth mentioning that ReinMax is not applicable to continuous random variables. While extending ReinMax to these cases is possible, it may not be necessary, given that many commonly used distributions can be reparameterized. For example, the normal distribution $z \sim \mathcal{N}(\mu, \sigma)$ can be re-written as $z = \mu + \sigma \cdot \mathcal{N} (0, 1)$, making it trivial to compute $\partial z/\partial \mu$ and $\partial z/\partial \sigma$.
>
> We hope our responses have adequately addressed your concerns and further highlighted the innovations and potential impact of our study. If you have any further questions or need additional information, please do not hesitate to ask.

---

> > ### Comment · Reviewer_GD4h · 2023-08-15
> > **Response to the Authors**
> >
> > I have read the other reviews and the authors' rebuttals and have decided to maintain my score. A majority of the reviewers feel that this paper should be accepted; the main contention is over the degree of significance. I hope that the authors include the new experiments, as well as the reviewers' suggestions, in the revised paper.

---

> > > ### Author Response · Authors · 2023-08-16
> > > **Thanks for your comments**
> > >
> > > Thank you for acknowledging our contribution. To highlight the significant potential of our proposed method, we provide additional experimental results (as elaborated in our general rebuttal). We will incorporate these discussions in our revised version.

---

### Author Rebuttal · Authors · 2023-08-10

# General Rebuttal

We thank all reviewers for their thoughtful feedback. In this work, we tackle critical challenges in gradient estimation for discrete variables, and our contributions are notable in two primary areas:
- we formally established that the straight-through estimator is a first-order approximation of the gradient,
- we proposed ReinMax, offering a second-order approximation with negligible computational overhead and consistent performance improvements.

In this general rebuttal, we provide additional experiment results on:
- Comparisons with the REINFORCE variant that employs the state-of-the-art variance reduction technology, namely RODEO (SHI 2022).
- Applications of ReinMax on real-world problems, specifically differentiable neural architecture search on CIFAR10, CIFAR100, and ImageNet-16-120.

## Comparisons with RODEO

### Bernoulli VAEs

We utilized ReinMax to train Bernoulli VAEs on MNIST, Fashion-MNIST, and Omniglot, adhering closely to the experimental settings of RODEO (SHI et al., 2022),  including pre-processing, model architecture, batch size, and training epochs. As summarized in Tables A and B, ReinMax consistently outperforms RODEO across all settings.

**Train A: -ELBO of 2 x 200 VAE on MNIST, Fashion-MNIST, and Omniglot when K=3 (i.e., three evaluations per image). \* Baseline results are referenced from SHI et al. (2022).**
||ARMS$^*$|DoubleCV$^*$|RODEO$^*$|RELAX$^*$|ReinMax|
|-|-|-|-|-|-|
|MNIST |100.84±0.14|100.94±0.09|100.46±0.13|101.99±0.04|97.83±0.36|
|Fashion-MNIST|237.05±0.12|237.40±0.11|236.88±0.12|237.74±0.12|234.53±0.42|
|Omniglot|115.32±0.07|115.06±0.12|115.01±0.05|115.70±0.08|107.51±0.42|

**Train B: -ELBO of 2 x 200 VAE on MNIST, Fashion-MNIST, and Omniglot when K=2 (i.e., two evaluations per image). \* Baseline results are referenced from SHI et al. (2022).**

||DisARM$^*$|Double CV$^*$|RODEO$^*$|ReinMax|
|-|-|-|-|-|
|MNIST |102.75±0.08|102.14±0.06|101.89±0.17|98.05±0.29|
|Fashion-MNIST|237.68±0.13|237.55±0.16|237.44±0.09|234.86±0.33|
|Omniglot|116.50±0.04|116.39±0.10|115.93±0.06|107.79±0.27|

### Polynomial Programming

To better understand the difference between RODEO and ReinMax, we conduct more experiments on polynomial programming, i.e., $\min_\theta E_{X} \Large[\normalsize \frac{\|X - c\|_p^p}{L}\Large]\normalsize$. Specifically, we consider polynomial programming under two different settings that define $c$ differently (the difference between these two settings is elaborated at the end of the part):
- Setting A: $c = [0.45, \cdots, 0.45]$. This is the setting we used in the submission.
- Setting B: $c = [\frac{0.5}{L}, \frac{1.5}{L}, \cdots, \frac{L-0.5}{L}]$.

We visualized the training curve of polynomial programming in the attached pdf (Figure 13), together with the training curve of Bernoulli VAE (Figure 12). ReinMax achieves better performance in more challenging scenarios, i.e., smaller batch size, more latent variables, or more complicated problems (Setting B or VAEs). Meanwhile, REINFORCE and RODEO achieve better performance on simpler problem settings, i.e., larger batch size, fewer latent variables, or simpler problems (Setting A). This observation matches our intuition:
- REIFORCE-style algorithms excel as they provide unbiased gradient estimation but may fall short in complex scenarios, since they only utilize the zero-order information, i.e., a scalar $f(\cdot)$ for each training instance.
- ReinMax, using more information (i.e., a vector $\frac{\partial f(D)}{\partial D}$ for each training instance), handles challenging scenarios better. Meanwhile, as a consequence of its estimation bias, ReinMax leads to slower convergence in some simple scenarios.

As to the difference between the Setting A and the Setting B, we would like to note:
- In Setting A, since $\forall i, c_i=0.45$ and $\theta_i\sim Uniform(-0.01, 0.01)$ at initialization, $E_{X_i \sim \mbox{softmax}(\theta_i)}\Large[\normalsize \frac{\|X_i - c_i\|_p^p}{L}\Large]\normalsize$ would have similar values. Therefore, the optimal control variates for $\theta_i$ are similar across different $i$.
- In Setting B, we set $c_i$ to different values for different $i$, and thus the optimal control variate for $\theta_i$ are different across different $i$.

Therefore, Setting A is a simpler setting for applying control variate to REINFORCE.

## Differentiable Neural Architecture Search

We demonstrate the applicability of ReinMax as a drop-in replacement in differentiable neural architecture search.

GDAS (Dong & Yang, 2019) is an algorithm that employs STGS to estimate the gradient of neural architecture parameters with a temperature schedule (decaying linearly from 10 to 0.1). We replaced STGS with ReinMax as the gradient approximator and changed the minimal temperature from 0.1 to 1.1(as discussed in Section 5 and Section 6.2, temperature scaling plays a different role in ReinMax).

We evaluate the resulting algorithm with the official implementation under the topology search setting in the NATS-Bench benchmark (Dong et al., 2020), and summarize the results in Table C as below. ReinMax brings consistent performance improvements over the baseline across all three datasets, demonstrating the great potential of ReinMax. We will add more analyses and discussions in the revision.

**Train C: Performance in NATS-Bench.\* Baseline results are referenced from Dong et al. (2020).**

||CIFAR-10 DEV|CIFAR-10 TEST|CIFAR-100 DEV|CIFAR-100 TEST|ImageNet-16-120|
|-|-|-|-|-|-|
|GDAS-Straight-Through Gumbel-Softmax$^*$|89.68±0.72|93.23±0.58|68.35±2.71|68.17±2.50|39.55±0.00|
|GDAS-ReiMax|89.92±0.27|93.47±0.35|69.40±1.63|69.61±1.71|41.11±2.09|

Dong, X. and Yang, Y. Searching for a robust neural architecture in four GPU hours. *CVPR*, 2019.

Dong, X., Liu, L., Musial, K., and Gabrys, B. Nats-bench: Benchmarking nas algorithms for 307 architecture topology and size. *TPAMI*, 2020.

---

### Author Response · Authors · 2023-08-21
**Thanks all reviewers for their valuable time, efforts, and constructive suggestions**

We would like to express our gratitude to the reviewers for their valuable time, efforts, and constructive suggestions.

To briefly recap our primary contributions:
1. We formally establish that Straight-Through works as a first-order approximation in the general multinomial case.
2. We propose a novel and sound gradient estimation method ReinMax that achieves second-order accuracy without requiring any second-order derivatives. ReinMax is shown to improve state-of-the-art methods in extensive experiments.

In our submission, we adhered to the experiment design of the existing study, focusing on small-scale problems for controllability and resource efficiency. In the discussion, we presented additional experiment results across diverse settings, including a real-world application, to further demonstrate the potential of ReinMax.

We also wish to note that we will utilize the extra space available in the final version to enhance our paper's clarity and presentation.

---

### Decision · Program_Chairs · 2023-09-21

**Decision:**

Accept (oral)

**Comment:**

All six reviewers agreed this paper should be accepted: it is original, high quality, clear, and provides (to the best of my knowledge) the first formal result showing the straight-through estimator is a first-order approximation in the multinomial case. Given that ReinMax has similar computational complexity to straight-through and improves performance it will likely gain traction as a drop-in replacement. This is a clear accept. Authors: you've already indicated that you've updated the submission to respond to reviewer changes, if you could double check their comments for any recommendation you may have missed on accident that would be great! The paper will make a great contribution to the conference!